# Centimeter-scale nanomechanical resonators with low dissipation

Andrea Cupertino[1,6], Dongil Shin ®[1,2,6], Leo Guo[3], Peter G. Steeneken ®[1,4], Miguel A. Bessa ®[5,7] ✉ & Richard A. Norte ®[1,4,7] ✉

High-aspect-ratio mechanical resonators are pivotal in precision sensing, from macroscopic gravitational wave detectors to nanoscale acoustics. However, fabrication challenges and high computational costs have limited the length-to-thickness ratio of these devices, leaving a largely unexplored regime in nano-engineering. We present nanomechanical resonators that extend centimeters in length yet retain nanometer thickness. We explore this expanded design space using an optimization approach which judiciously employs fast millimeter-scale simulations to steer the more computationally intensive centimeter-scale design optimization. By employing delicate nanofabrication techniques, our approach ensures high-yield realization, experimentally confirming room-temperature quality factors close to theoretical predictions. The synergy between nanofabrication, design optimization guided by machine learning, and precision engineering opens a solid-state path to room-temperature quality factors approaching 10 billion at kilohertz mechanical frequencies – comparable to the performance of leading cryogenic resonators and levitated nanospheres, even under significantly less stringent temperature and vacuum conditions.

Mechanical resonators are crucial in precision sensing, enabling gravitational-wave observations at the macroscale[1,2], probing weak forces in atomic force microscopy at the nanoscale[3–7], or playing a central role in recent quantum technologies[8–10]. Their performance largely hinges on having low mechanical dissipation, quantified by the mechanical quality factor ($Q$). The $Q$ factor measures both radiated acoustic energy and infiltrating thermomechanical noise, with a high $Q$ pivotal in preserving resonator coherence. This is crucial, particularly at room temperature, for observing quantum phenomena[11,12], advancing quantum technology[13], and maximizing sensitivity for detecting changes in mass[14–16], force[17,18], and displacement[1]. High $Q$ is often achieved through "dissipation dilution", a phenomenon originating from the synergistic effects of large tensile stress and high-aspect-

ratio, observed both in resonators with macroscopic lengths on the order of centimeters and above[2,19] and resonators with nanometers thicknesses[20–22]. The reader is referred to Cagnoli et al.[19] for a discussion about the use of 'dissipation dilution' terminology for pendula and prestressed nanomechanical resonators.

At the macroscale, a notable example is the pendulum formed by the mirror-suspension pair in gravitation wave detectors[23,24], where kilogram mirrors are suspended by wires tens of centimeters long and a thickness on the order of micrometers (Fig. 1b). The high tensile stress is created by the mirror masses, which induce stress in the wires due to gravity. The resulting high-quality factor of the pendulum modes of the order of $10^8$ allows for isolating the detector from thermomechanical noise, enabling it to reach its enhanced displacement

[1]Department of Precision and Microsystems Engineering, Delft University of Technology, Mekelweg 2, 2628 CD Delft, The Netherlands. [2]Department of Materials Science and Engineering, Delft University of Technology, Mekelweg 2, 2628 CD Delft, The Netherlands. [3]Department of Microelectronics, Delft University of Technology, Mekelweg 2, 2628 CD Delft, The Netherlands. [4]Department of Quantum Nanoscience, Kavli Institute of Nanoscience, Delft University of Technology, Lorentzweg 1, 2628 CJ Delft, The Netherlands. [5]School of Engineering, Brown University, 184 Hope St., Providence, RI 02912, USA. [6]These authors contributed equally: Andrea Cupertino, Dongil Shin. [7]These authors jointly supervised this work: Miguel A. Bessa, Richard A. Norte. ✉e-mail: miguel_bessa@brown.edu; r.a.norte@tudelft.nl

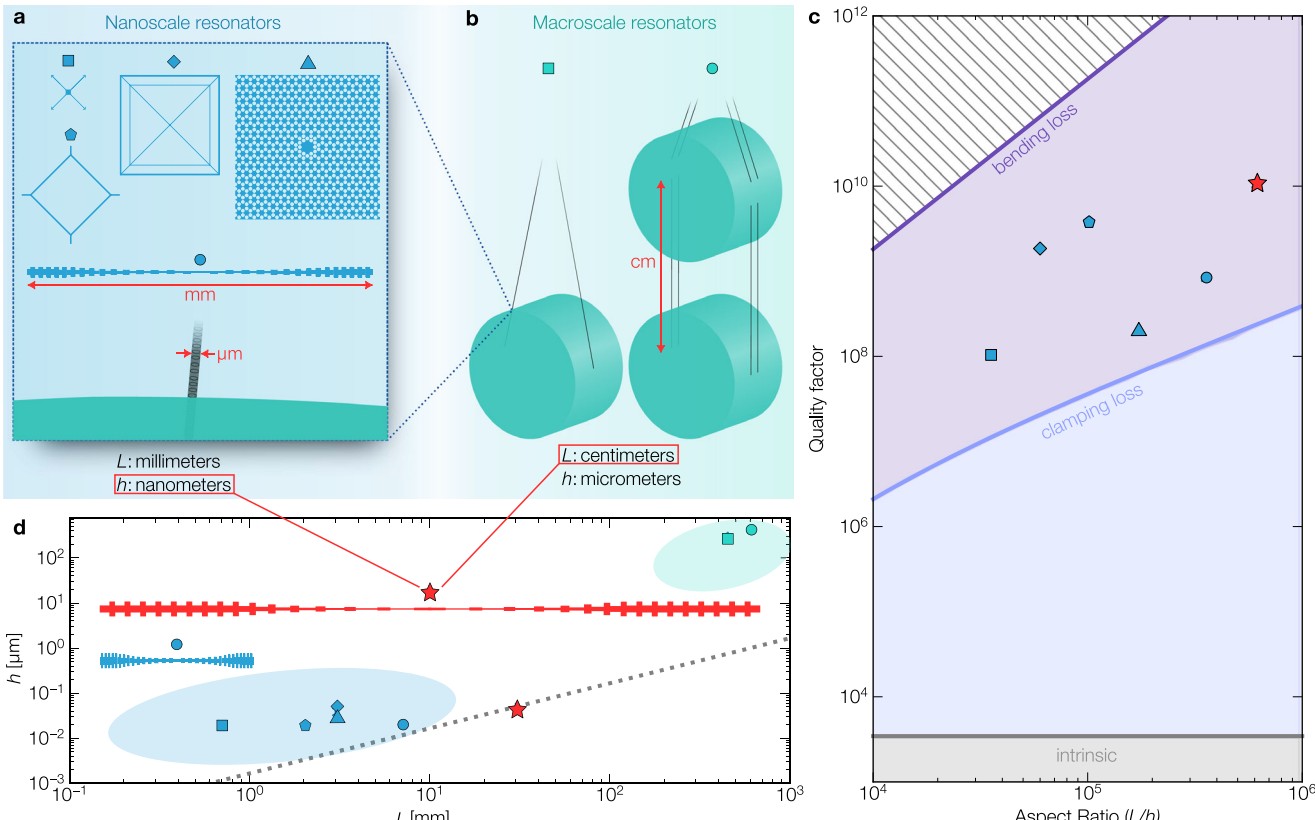

**Fig. 1 | High-aspect-ratio mechanical resonators. a** State-of-the-art Si₃N₄ nano-mechanical resonators with nanometers thickness and length below 1 cm. On the left side: trampoline resonator[30] and perimeter resonator[33]. At the center: spi-derweb resonator[34]. On the right side: soft-clamped membrane[35]. On the bottom: tapered string with phononic crystal[36] and a zoom-in of the LIGO suspension in **b** scaled to match the dimension of all the nanomechanical resonators. **b** Macromechanical resonators with micrometers thickness and extending into tens of centimeters. From left to right: LIGO suspension[24], advanced LIGO suspension[23]. **c** Predicted quality factor as a function of the mechanical resonators aspect-ratio for a Si₃N₄ string. The blue points indicate the measured quality factor for the devices in (**a**). The red stars show the predicted quality factor of the design demonstrated here. **d** Thickness versus length of state-of-the-art mechanical resonators in (**a** and **b**). The red star shows the design demonstrated here with its aspect-ratio indicated by the dotted black line.

sensitivity. The exceptional acoustic isolation of these resonators has enabled landmark demonstrations of quantum effects at an unprece-dented kilogram scale[25] and contributed to the first observations of gravitational waves[26]. Similar configurations at smaller scales are employed on table-top experiments to investigate the limits of quan-tum mechanics[27] and its interplay with gravity[28]. To date, resonators in this category, which span lengths of several centimeters, have been limited to minimum thicknesses on the order of micrometers. These types of mechanical resonators have been particularly successful in demonstrating low dissipation in room-temperature environments.

At the nanoscale, nanomechanical resonators possess sig-nificantly reduced thickness, in the range of sub-micrometers, with a length limited to a sub-millimeter length by convention. They are generally fabricated on top of a supporting substrate by standard silicon technology and integrated onto a single chip (Fig. 1a). High tensile stress, generated through a thermal mismatch between the resonator and the substrate, offers both lower mechanical dissipation and structural stability for long-distance suspended nanostructures. This necessitates careful material selection. Silicon nitride (Si₃N₄) is emerging as one of the most common and easily manufacturable materials. It has favorable optical and mechanical properties at room temperature and specific advantages gained through strained config-urations on silicon, emphasizing the role of strain engineering for dissipation characteristics. Leveraging standard silicon technology compatibility, these nanomechanical resonators provide scalability[29] and integration[8] not found in their macroscale counterparts. Techni-ques like mode-shape engineering[30–34] and phononic crystal (PnC)

engineering[35,36] have further enhanced dissipation dilution, pushing quality factors above 10⁹. This improved coherence of motion has propelled advances in quantum phenomena exploration at cryogenic temperatures[37,38] and fostered an advanced sensor class development[39,40]. Generally, the quality factor in this class of resona-tors (Fig. 1d) is proportional to the length, and the resonance fre-quency is inversely proportional to the length[21,34]. As nanoscale resonators become longer, their manufacturing yield drops dramati-cally due to compounding factors, including limits in devices per chip, the alignment of nanoscale features over centimeters, and as resona-tors become longer, they become increasingly delicate suspended structures. These combined decrease device yield and experimental match with expected Qs as resonators become more susceptible to small forces during nanofabrication. These overlapping fabrication challenges have made reliably increasing the length of nanoscale resonators beyond millimeter scales prohibitively difficult in practice.

Advancing these resonators to multi-centimeter lengths while maintaining their nanoscale thickness would uniquely combine the benefits of macroscale and nanoscale mechanical resonators and open an undiscovered regime for acoustic technologies. These suspended structures will be characterized by their ultrahigh room-temperature Q factors, their ability to firmly integrate with microchip architectures, and their relatively large masses and surface areas. Such centimeter-scale surface areas and masses, which are mechanically well-isolated, are well suited for high precision measurements of acceleration[41], pressure[42], and vacuum[43]. These attributes also make them promising for the observation of mesoscopic quantum behavior in ambient

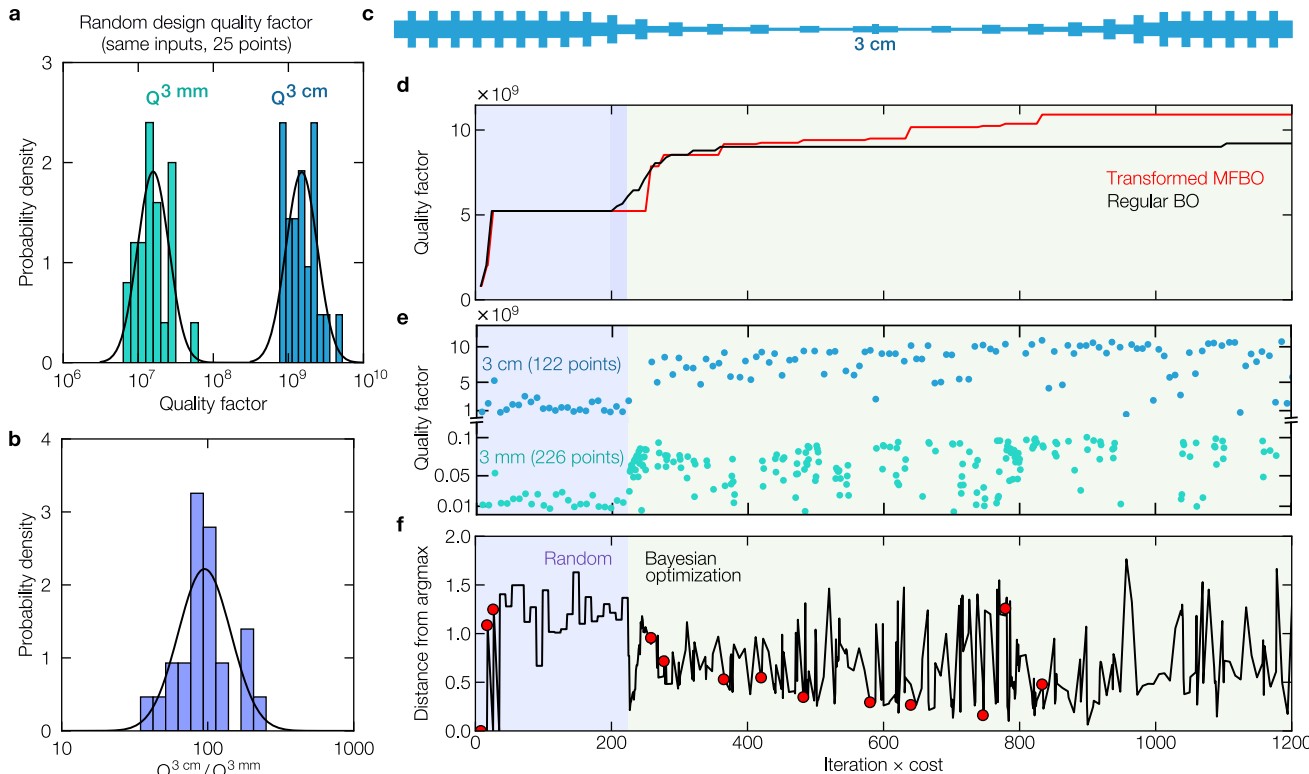

**Fig. 2 | Multi-fidelity Bayesian optimization. a** Quality factor distribution for the two different lengths of PnC resonators (Blue: 3 cm, Green: 3 mm). 25 sets of design parameters were randomly selected, and both results followed a log-normal distribution. **b** Probability distribution of the 3 cm and 3 mm resonator's quality factor ratio. The quality factor ratio follows a log-normal distribution. **c** Shape of the optimized 3 cm resonator. **d** Evolution of the optimized quality factor with two formulations. MFBO maximizing the log of the quality factor (Transformed MFBO) and single-fidelity Bayesian optimization maximizing the quality factor directly (Regular BO). Transformed MFBO outperformed regular BO. **e** Iteration history of the transformed MFBO where the shaded blue part represents the first 25 randomly selected simulations for both high and low-fidelity models, and where the green

part represents the remaining design iterations where the MFBO method searches the best design and controls what fidelity it wants to evaluate. The abscissa of the plot has units of iteration cost, and its limit was set to 1200. At the end of the optimization process, 122 high-fidelity simulations (each with a relative cost of 8) and 226 low-fidelity simulations (relative cost of 1) were evaluated. All the designs considered during the optimization can be found in the Supplementary Video. **f** The distance from a previous optimized point to the point considered in that iteration. The red markers indicate when the algorithm has found a higher quality factor until that iteration. **d**–**f** Shares the same *x*-axis corresponding to the design iteration times the cost of each iteration as the optimization process evolves.

temperatures, which are largely limited by room-temperature thermomechanical noise[44]. Reaching a quantum-limited motion regime at room temperature will extend the application range of quantum sensors[13], quantum calibrated sensing[45], and quantum memories[46]. The drive towards multi-centimeter nanostructures also holds the potential for applications in ultrasensitive nanomechanical detection[47], including searches for dark matter[48], Casimir forces[49,50], and studies of entropy and time[51]. Yet, pursuing such high-aspect ratios at the envisioned centimeter scale faces considerable computational and fabrication barriers, including the need for a high fabrication yield to offset the limited number of devices per chip and cost implications. This challenge of cost-effective high-yield fabrication for centimeter-scale nanostructures introduces a specific dynamic between economics and design distinct from traditional high-volume nanotechnology.

We propose an approach that merges delicate fabrication techniques and multi-fidelity Bayesian optimization, enabling the creation of mechanical resonators with centimeter lengths and high-aspect-ratios exceeding $4.3 \times 10^5$; equivalent to reliably producing ceramic structures with a thickness of 1 millimeter, suspended over nearly half a kilometer. Our design strategy curtails optimization time while successfully circumventing common fabrication issues such as stiction, collapse, and fracture. Applying our strategy to a 3 cm long silicon nitride string, we achieve a quality factor exceeding $6.5 \times 10^9$ at room temperature—the highest ever recorded for a mechanically clamped resonator. This solid-state platform performs on par with counterparts

such as levitated nanospheres, which require significantly more stringent vacuum conditions as low as $10^{-11}$ mbar[52] to reduce the collision rate between the particle and background gas molecules, typically the dominant source of dissipation. In contrast, our clamped centimeter-scale resonators are limited by intrinsic losses[20] in view of their higher vibrational frequency and can approach comparable quality factors at pressures nearly two orders of magnitude higher. Notably, our room-temperature resonators can also work at quality factors only observed in cryogenic counterparts[53,54]. This enhanced capability to operate at higher temperatures and pressures unveils the potential in centimeter-scale nanotechnology, expanding the boundaries of what is achievable with on-chip, room-temperature resonators.

## Results

### High-aspect-ratio advantage and multi-fidelity design

The quality factor of string resonators with a constant cross-section is given by[21]

$$Q = Q_{\text{int}} \left[ n^2 \pi^2 \frac{E}{12\sigma} \left( \frac{h}{L} \right)^2 + 2\sqrt{\frac{E}{12\sigma}} \left( \frac{h}{L} \right) \right]^{-1} \quad (1)$$

where $Q_{\text{int}}$ is the intrinsic quality factor (surface loss) that varies linearly with the resonator's thickness, $n$ is the mode order, $E$ is Young's modulus, $\sigma$ is the initial stress, $L$ and $h$ are the resonator's length and

thickness in the direction of motion, respectively. Note that this equation assumes perfect clamping to the substrate without considering the mechanical coupling, potentially affecting the $Q$ factor at a smaller mode order[55]. Since $Q_{int} \propto h$, increasing the length can be a more effective strategy (than reducing thickness) to increase quality factors. For given aspect ratio $h/L$, the first and second term in the denominator of Eq. (1) originates from bending loss and clamping loss, respectively, for which we use the definitions from Schmid et al.[21]. The clamping loss originates from the sharp curvature at the clamps where the resonator is anchored to the supporting substrate, and the bending loss accounts for the curvature along the remainder of the resonator, both depending on the aspect-ratio (Fig. 1c). The clamping region is defined as the part of the resonator at a distance of less than $L_c = \sqrt{E/6\sigma}h$ from the clamping points where the mode shape deviates strongly from a simple sine function[21]. The curvature is defined as the reciprocal of the radius of curvature for the resonator's bending deformation.

In the high-aspect-ratio limit, the contribution of clamping loss is orders of magnitude larger, dominating the loss contribution. To this end, techniques known as soft clamping have been employed to reduce the sharp curvature at this $L_c$ region[36] and improve the $Q$ factor so that the $Q$ factor scales quadratically with the aspect-ratio of the resonator. Those include phononic crystal-based string and membrane resonators, which employ a higher-order eigenmode confined in a central defect by the acoustic bandgap[35,36], spiderweb and perimeters resonators, which exploit low-order eigenmodes[33,34], or other methods considering the fundamental mode[30,31] (Fig. 1a). It is important to notice that Eq. (1) is a straight beam formulation that does not directly apply to phononic crystal strings[56] (due to the geometry difference) but can give insight into general trends with $h/L$. A high aspect-ratio is beneficial even if the $Q$ factor is limited by the curvature at the clamping points (Eq. (1)). Hence, state-of-the-art high $Q$ factor nanoresonators have advanced toward devices with increasing aspect-ratios, pushing the total length from micrometers[30,57] to millimeters[33–36] (Fig. 1c) alongside a thickness reduction below 50 nm.

Centimeter-long resonators maintaining a thickness within tens of nanometers can significantly raise the achievable quality factor. However, the resulting higher aspect-ratios demand advanced simulation-based design strategies due to the elevated computational cost of accurately capturing their behavior through direct simulation models. Numerical analyses require increased degrees of freedom (DOFs) to describe models with extreme aspect ratios. For instance, conducting finite element analysis on a centimeter-long string with phononic crystals requires roughly ten times more DOFs than the same geometry with a one-order lower aspect-ratio. This poses a challenge for centimeter-scale designs, as the heightened computational cost sets a practical boundary, particularly given the limited availability of high-fidelity data. One optimization process took about 16 CPU-days for 150 iterations, which created a bottleneck in the design process. Here, multi-fidelity Bayesian optimization (MFBO)[58,59] alternates between employing both a quick, low-fidelity model for 3 mm resonators and a slower, high-fidelity model for 3 cm ones. The high-fidelity predictions were eight times slower on average. We utilize finite element simulations with COMSOL[60] to maximize the quality factor. Despite the existence of analytical derivations for one-dimensional beams[56], we discuss direct numerical simulations in this study. This choice aims to provide a more generalized design approach for large aspect-ratio resonators. The approach in this study could be expanded for other high $Q$ resonators sensitive to the length scale but lacking derived analytical formulations[33–35]. The geometry of both resonators[36] is parameterized (Supplementary Information Section 9) to allow the presence of a PnC with a defect embedded in the center and chosen to practically allow the most number of on-chip resonators. The method then quickly explores the design space by fast evaluations of the smaller structures at higher frequencies (MHz) while establishing a correlation with the large devices at lower frequencies. This enables it to probe (slow) solutions for the high-aspect-ratio structures only on rare occasions when it expects the design to achieve a large quality factor.

The effectiveness of MFBO relies on the correlation between low and high-fidelity models and their respective evaluation times. If the low-fidelity model lacks correlation with the high-fidelity model, or if its evaluation time is comparable, the method loses effectiveness. By parameterizing the design space independently from resonator length (see the "Methods" section), we observed a reasonable correlation between the 3 mm (low-fidelity) and the 3 cm (high-fidelity) resonator designs (Fig. 2a). The $Q$ factor of the 3 cm design was a hundred times larger on average, confirming the expected correlation between the two models according to the soft clamped $Q$ factor in Eq. (1). The variation in Fig. 2b underscores that what improves performance for a 3 mm resonator does not necessarily translate into a better 3 cm resonator. Therefore, exclusively relying on the low-fidelity model (3 mm resonator) would not yield the desired outcome for designing the 3 cm resonator. Simultaneously, despite each resonator size having its unique set of optimal geometric parameters, the algorithm is still capable of learning enough from the response of the 3 mm resonator to guide the optimization of the 3 cm one. Consequently, we applied MFBO to the log-normal $Q$ factor, letting the algorithm selectively probe the design space via whichever fidelity it chooses. In essence, MFBO uses millimeter-scale simulations to guide centimeter-scale optimization.

Even though increasing the length of the resonator has a trend towards increasing the resonator's quality factor, optimization of the phononic crystal parameters is required for each given specific length scale. The advantage of using multi-fidelity transformed Bayesian optimization is shown in Fig. 2d, as it finds a design with $Q$ outperforming single-fidelity Bayesian optimization (regular BO) $Q$ by ~20%. Figure 2c depicts the optimized geometry. After determining the optimal design, we focused on fabricating this high-aspect-ratio device. The optimized geometry follows a tapering phononic crystal shape, which is similar to the result suggested by Ghadimi et al.[36]. The optimal design maximized the unit cell width around the clamping region and narrowed down the width of the unit cells coming near the center of the resonator. One distinguishing aspect of our approach was that during the optimization under the design domain, the algorithm considered both symmetric and anti-symmetric modes, but the optimized mode was selected to be anti-symmetric rather than symmetric. Because of this difference, the defect width was not minimized, unlike the early study[36], since the kinetic energy with significant movement is not maximized at the center. All the designs considered during the optimization can be found in the Supplementary Video. Once the best design was found, we focused on addressing the challenges of fabricating a device with such an extreme-aspect-ratio.

## Centimeter scale nanofabrication

Manufacturing centimeter-scale nanoresonators relies on fabrication intuition that shifts from conventional nanotechnologies in design principles, fabrication methods, and cost considerations. In accordance with Moore's Law, conventional nanotechnology has focused on miniaturization across all three dimensions ($x$, $y$, $z$). However, centimeter-scale nanotechnology marks a transition that requires components expanding out to macroscopic lengths in $x$ and/or $y$ while retaining their nanoscale thickness. These nanostructures not only have high-aspect-ratios and centimeters length at the macroscale, but they are also patterned at the nanoscale with small feature sizes, providing them with enhanced functionalities (mechanical, optical, etc.). In particular, our nanostrings are not just elongated and thin; they also incorporate precisely patterned phononic bandgaps.

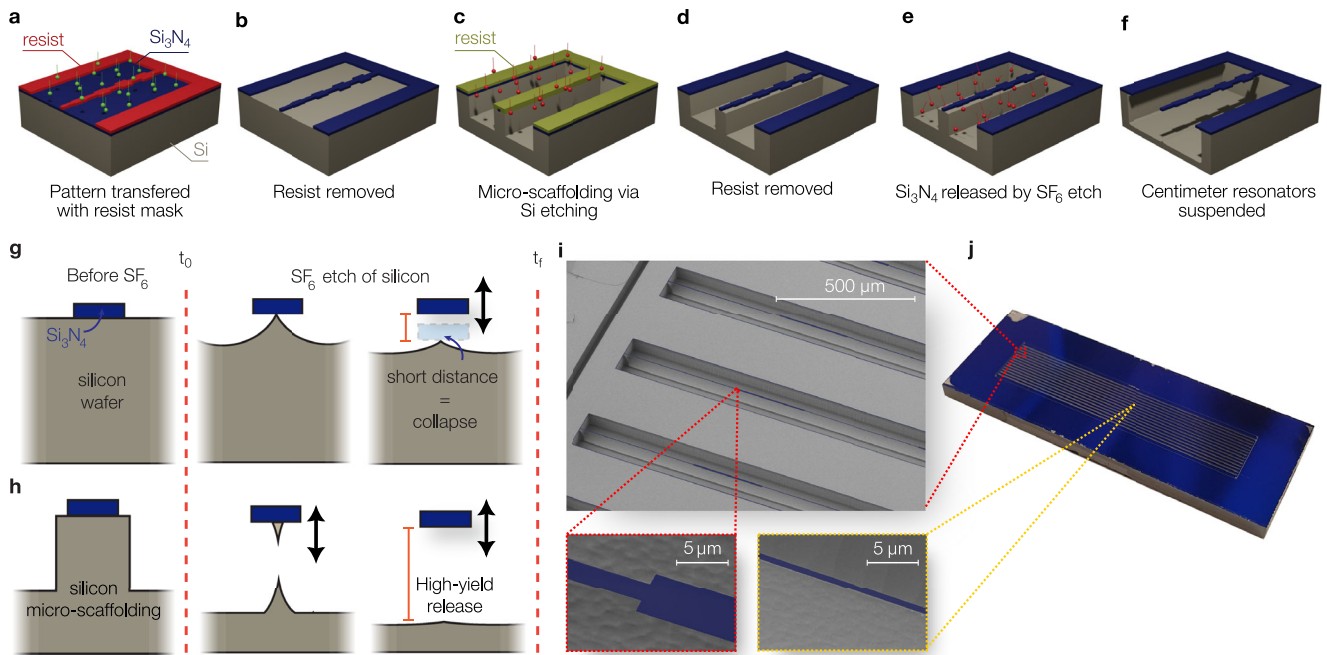

**Fig. 3 | Fabrication of centimeter-scale nanoresonators. a–f** Schematic of the fabrication process composed of $Si_3N_4$ patterning via dry etching (**a**), mask removal (**b**), cryogenic DRIE silicon etching with photoresist mask (**c**), mask removal (**d**), $Si_3N_4$ undercut (**e**), and $Si_3N_4$ suspended over a large gap (**f**). **g** $SF_6$ release step when the $Si_3N_4$ structure is suspended after being patterned, leading to collapse due to the short distance with the supporting silicon wafer. **h** $SF_6$ release step with micro-scaffolding when the $Si_3N_4$ structure is suspended after a DRIE of the supporting silicon wafer, resulting in a large gap and high-yield release. **i** False-colored scanning electron microscope pictures of the optimized 3 cm nanoresonators at the clamping area (top), at the boundary of the first unit cell (bottom-left), and at the center (bottom-right). The blue area indicates the area where $Si_3N_4$ is suspended, consisting of the string and the overhang. **j** Photograph of a chip containing fourteen centimeter-scale nanoresonators, each 3 cm long.

In contrast to conventional miniaturization, the size constraints of centimeter-scale nanoresonators permit far fewer devices per wafer, requiring high fabrication yields due to the resulting higher costs per device. Given their long geometries, any fracture of a centimeter-scale nanoresonator not only results in fewer successful devices, but these broken devices can also collapse over several neighboring structures, escalating the implications of fabrication errors and low yield. Moreover, the fabrication of these high-aspect-ratio structures requires them to be released with delicate nanofabrication techniques that do not exert any destructive forces during and after suspension. These processes must ensure that the high-aspect-ratio structures remain unfractured and undistorted and are positioned safely away from nearby surfaces to prevent potential issues such as stiction due to attractive surface forces.

The schematic of the fabrication procedure is shown in Fig. 3a–f (see the "Methods" section). First, high-stress (1.07 GPa) $Si_3N_4$ is deposited on a silicon wafer. A resist layer is spun on and lithographically patterned with an array of the 3 cm long optimized PnC resonators. This is achieved by employing multiple exposures with varying resolutions to reduce the exposure time without compromising the accuracy (Supplementary Information Section 3) This patterned resist is used as a mask to transfer the design into the $Si_3N_4$ film via directional plasma etch (Fig. 3a). Typically the most crucial step is the careful release of these fragile structures by removing the silicon beneath the $Si_3N_4$ resonator; with centimeter-scale nanostructures the requirements for successful suspension become much more stringent.

A critical aspect of realizing these nanostructures is the high stress within $Si_3N_4$, which not only contributes to achieving a high $Q$ factor but also provides the required structural support and stability, allowing these taut strings to remain free-standing over remarkable distances without collapsing or sagging. In particular, we use dry $SF_6$ plasma etch to remove the silicon under the $Si_3N_4$ resonator (Fig. 3c–f) since it avoids the conventional stiction and collapse from surface tension present in liquid etchants[61] and it does not leave any residues. Once released, these 3 cm-long, 70 nm-thick structures can displace tens of micrometers due to handling and static charge build-up and potentially collapse onto nearby surfaces. In combination with the $SF_6$ plasma dry release, we first engineer a micro-scaffolding[36] (Fig. 3g, h) into the silicon underneath our $Si_3N_4$ that allows the free-standing structures to be suspended quickly, delicately and far away from the substrate below, significantly increasing the yield and viability of this proposed nanotechnology (Supplementary Information Section 5). The micro-scaffolding is lithographically defined by a second exposure and transferred into the silicon substrate via deep directional plasma etch (Fig. 3c). While our simulated strings are designed with a nominal thickness of 50 nm, the fabricated resonators have a larger final thickness from edge to center along their length. This is caused by the difficulty of dissipating heat during the etching step of the silicon substrate, which is expected to decrease the $Q$ factor predicted from the simulation (Supplementary Information Section 2).

Our choice to focus on 1D PnC nanostrings comes from a practical standpoint, which allows for packing numerous devices per chip (Fig. 3i, j). By carefully engineering the delicate release of these structures, we achieved a fabrication yield as high as 93% on our best chip and 75% overall chips processed. While we only study 1D structures, the methodologies we developed are versatile and can be readily applied to more complex structures such as 2D phononic shields[35].

## Low dissipation at room temperature

To assess the mechanical properties of the fabricated nanoresonators, we characterized the strings' out-of-plane displacements by a balanced homodyne optical interferometer (Fig. 4c) built to experimentally measure the 3 cm nanoresonators. The nanoresonator is placed inside an ultra-high vacuum (UHV) chamber at $P < 10^{-9}$ mbar to avoid gas

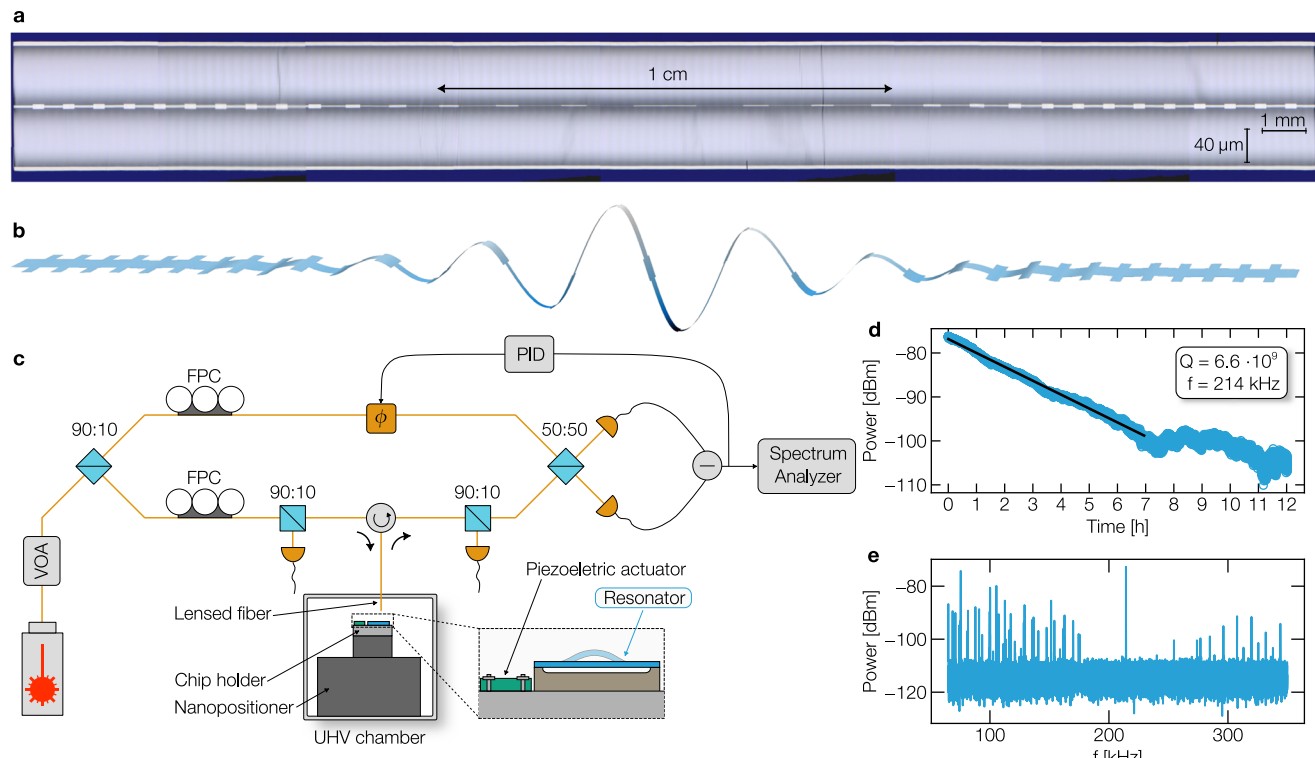

**Fig. 4 | Quality factor validation. a** Optical picture of the optimized 3 cm nanor-esonators. For illustration purposes, the picture is scaled along the vertical direction as indicated by the two scale bars. **b** Mode shape of the eigenmode predicted to have high quality factor extracted from finite elements analysis. **c** Balanced homodyne optical interferometer built to experimentally measure the 3 cm nanoresonators. The nanoresonator is placed inside an ultra-high vacuum (UHV) chamber and mechanically excited by a piezoelectric actuator. Its motion is detected by an infrared laser via a lensed fiber. VOA variable optical attenuator, FPC fiber polarization controller, PID proportional integral derivative controller, Φ fiber stretcher. **d** Ringdown trace of optimized nanoresonator excited at 214 kHz. The black solid line indicates the linear fitting corresponding to an extracted quality factor of 6.6 billion at room temperature. **e** Thermomechanical noise spectrum.

damping, increasingly dominant for high-aspect-ratio structures (see Supplementary Information Section 1), and mechanically excited by a piezoelectric actuator. The resonator's motion is detected by an infrared laser via a lensed fiber. Figure 4e shows the displacement spectrum obtained from a location near the center of the string, under thermal excitation, for the device depicted in Fig. 4a. The spectrum shows a clear bandgap in frequency between 175 and 255 kHz with one localized mode inside. The latter is observed at 214 kHz, in good agreement with simulation prediction. Figure 4b shows the predicted mode shapes obtained by finite element analysis for the eigenmode in the center of the bandgap. On the contrary, outside the 175–255 kHz range, a plethora of modes are detected, whose displacement is distributed over the entire string length.

We experimentally evaluate the quality factor of the nanor-esonators by applying a sinusoidal function to the piezoelectric actuator at a frequency near the eigenfrequency of the localized mode. Once the displacement at resonance reaches a plateau, the excitation is abruptly turned off to measure the ringdown of the mechanical mode. Figure 4d shows the envelope of the obtained signal for our best-performing device, where the measured decay rate is proportional to the nanoresonator energy dissipation and, thus, its quality factor. As Fig. 4d shows, the localized mode at 214 kHz decays for over 7 h. This corresponds to a $Q$ factor of 6.6 billion at room temperature. To corroborate these findings, we tested additional lithographically identical devices, exhibiting the same frequency response in good agreement with the simulation and quality factors within a 50% range (Supplementary Information Section 10).

While our simulations expect $Q$s of 10 billion, the fabricated centimeter scale nanomechanical structures have high-aspect-ratios that make it difficult to dissipate heat during the (exothermic) undercut process (Fig. 3e), resulting in different $Si_3N_4$ etch rates. This

gives the nanostructures slightly different thicknesses and dimensions from edge to center along the beam, thus reducing the fidelity between design and experiment with a lower measured $Q$ (Supplementary Information Section 2).

## Discussion

We demonstrated centimeter-scale nanomechanical resonators with aspect-ratio above $4.3 \times 10^5$. Our approach combines two features. First, using MFBO, we are able to reduce the simulation cost while maintaining the required accuracy to precisely capture the resonator's behavior. The resulting data-driven design process allows the optimization of PnC strings, obtaining soft-clamped modes that eliminate clamping losses and radiation to the substrate. Second, a dry etching technique that overcomes limitations such as stiction and collapse enables us to reliably realize the optimized designs on-chip. The fabricated PnC strings extend for 3 cm in length maintaining nanometers thickness and a minimum width of 500 nm. With a $Q$ of 6.6 billion at a frequency of 214 kHz, we experimentally achieved the highest $Q$ factor yet measured for clamped resonators at room temperature.

The obtained aspect-ratio enables not only to achieve a two-fold improvement of the quality factor in room-temperature environments, but it also leads to low resonance frequencies and large spacing between nearby mechanical modes. Those features translate into coherence time $t_{coh} = \hbar Q / (k_B T)$ approaching 1 ms and thermomechanical-limited force sensitivity of $\sqrt{4 k_B T \omega_m m / Q} \sim$ aN Hz$^{-0.5}$.

A natural application is ground-state cooling of the mechanical resonators in a room-temperature environment. The high coherence time enables these resonators to undergo more than $Q \omega_m (\hbar / K_B T) \approx 200$ coherent oscillations in the ground state before a thermal phonon enters the system[62]. The small thermal decoherence

rate $\Gamma_{th} = k_B T/(\hbar Q)$ of a few Hz allows to resolve the zero point fluctuations $x_{zp} = \sqrt{\hbar/(2m\omega_m)}$ with a displacement imprecision $S_{xx}^{imp} = 4x_{zp}^2/\Gamma_{th}$ below $1 \times 10^{-15}$ m Hz$^{0.5}$, on pair with the imprecision limit due to shot noise achievable in conventional interferometer setup. This makes centimeter-scale nanoresonators particularly promising for the cavity-free cooling scheme[63]. The developed resonators are also ideal candidates for creating high-precision sensors, specifically force detectors[18], and hold promise for obtaining frequency stability in pairs with state-of-the-art clocks[64–66]. Conservatively assuming sub-micrometers amplitude displacements in the linear regime, we can extrapolate a thermomechanical limited Allan deviation[65] $\sigma_y(\tau) \sim 3 \times 10^{-12}/\sqrt{\tau}$ for $m_{eff} = 4.96 \times 10^{-13}$ kg.

Remarkably, the degree of acoustic isolation (quantified by the quality factor) we can achieve on a solid-state microchip is similar to values recently demonstrated for levitated nanoparticles operating at vacuum pressure levels more than two orders of magnitude lower than our resonators[13,52]. This comparison gains further importance when considering that levitated particles are essentially isolated from the environment, interacting only minimally with residual gas molecules at vacuum levels as low as $10^{-11}$ mbar, which requires infrastructures different from the turbo and roughing pump combination used in this study (Supplementary Information Section 11). In contrast, our solid-state resonators are physically clamped to a room-temperature chip, surrounded by 100 times higher gas pressures, and exhibit comparable acoustic dissipation.

The ability to combine macroscale resonators with nanomechanics offers unique possibilities to integrate the versatility of on-chip technology with the detection sensitivity of macroscale resonators. Notably, the only foreseeable limitation to producing even longer, higher-$Q$ devices is that larger undercut distance must be engineered, and practically going to longer, lower-frequency devices would require increasingly higher vacuum levels; this makes centimeter-scale nanotechnology particularly interesting for next-generation space applications[67,68], which inherently operate at pressures below $10^{-9}$ mbar. Pushing the boundaries of fabrication capabilities with higher selectivity materials[53,69,70] would extend our current approach to more extreme-aspect-ratios and investigate unhackneyed physics. These include the exploration of weak forces such as ultralight dark matter[48] and the investigation of gravitational effects at the nanoscale[28,71]. While Si$_3$N$_4$ resonators have demonstrated the highest room-temperature quality factors, it is noteworthy that the quality factor for Si$_3$N$_4$ resonators has been consistently reported to increase at cryogenic temperatures[72,73]. Based on conservative estimates, we predict that our centimeter-scale resonators could exhibit $Q$ factors above $6 \times 10^{10}$ at cryogenic temperatures, potentially surpassing current cryogenic devices[53,54]. By blurring the line between macroscopic and nanoscale objects, these centimeter-scale nanomechanical systems challenge our conventional intuitions about fabrication, costs, and computer design and promise to give us innovative capabilities that have not been available at smaller scales.

## Methods
### Computational experiments and design
The design approach for high-aspect-ratio resonators was based on numerical analysis with COMSOL[60]. The quality factor was maximized via multi-fidelity Bayesian optimization without recurring to the analytical solution derived for beam-like phononic crystals (PnCs)[56]. In particular, we consider the trace-aware knowledge gradient (taKG) formulation of Bayesian optimization with two fidelities[74]. Detailed information about the formulation can be found in Supplementary Information Section 7. The maximization based on the high-fidelity model becomes possible by learning the trend (surrogate model) from multiple low-fidelity predictions instead of using fewer high-fidelity evaluations. The approach is especially beneficial for cases when the

difference in time evaluation between fidelities is significant, i.e., the time it takes to perform one function evaluation (one design prediction via COMSOL) for the high fidelity is much longer than for the lower fidelity.

We considered a 3 cm resonator as the high-fidelity model and a 3 mm resonator as the low-fidelity model for the MFBO. As mentioned in the main text, we expected them to be correlated, given that $Q \propto L^2$ (Eq. (1)) for the string type resonators neglecting the sharp curvature change around the clamping region using the PnC. This correlation allows us to predict the response of the 3 cm computationally expensive model by the 3 mm relatively cheap model. For the centimeter-scale PnC resonator's quality factor maximization, we designed the resonator's geometry with a two-dimensional model. The model has nine design parameters, including five determining the resonator's overall shape, the unit cell's width and length ratio, and the defect's length and width. Design variables were set to be independent of the resonator length. Detailed parameter descriptions can be found in Supplementary Information Section 9.

Figure 2a shows the quality factor distribution obtained from randomly selected 25 high-fidelity PnC resonator designs and the same number for low-fidelity ones. For both lengths, the same design parameters are considered. Both length scale's quality factor follows a log-normal distribution. More importantly, the ratio between the two fidelities also follows a log-normal distribution as depicted in the histogram in Fig. 2b. The result indicates that the $Q$ factor of the 3 cm design is, on average, a hundred times larger than that of the same design, scaled down to 3 mm, which confirms the expected correlation between the two models. Nevertheless, the ratio shows significant variance, ensuring that the optimum design for the 3 mm resonator does not precisely correspond to the best design for the 3 cm case. These findings underscore that using MFBO with low- and high-fidelity simulation models leads to a balance between obtaining the required accuracy and minimizing the simulation cost.

Figure 2d–f shows the optimization iteration history, considering that the high-fidelity simulation costs eight times more than the low-fidelity simulation. This average time difference between fidelities is determined from the initial designs obtained by random search. Figure 2d compares the results when the logarithm of the quality factor is considered for the maximization using multi-fidelity Bayesian optimization (transformed MFBO) and when single-fidelity Bayesian optimization is directly optimizing the quality factor (regular BO). After starting with 25 randomly selected initial calculations for the 3 cm and 3 mm model (or only the 3 cm model for the regular BO), the algorithm maximizes the quality factor for the 3 cm model by searching for the best possible design parameters. We note that the initial random design of experiments affects the optimization performance, but in most cases, MFBO outperformed single-fidelity Bayesian optimization. The results with different random initials comparing the transformed MFBO and regular BO are summarized in Supplementary Information Section 8.

Detailed information on the transformed MFBO is shown in Fig. 2e and f. Figure 2e illustrates the quality factor calculated for each fidelity, and Fig. 2f illustrates the distance from a previous optimized point to the point considered in that iteration. Right after the random search, the algorithm runs predominantly low-fidelity simulations to optimize the quality factor, taking advantage of the relatively cheap simulation cost. We note that the low-fidelity simulation has higher quality factor variance when compared to the high-fidelity simulations, given the larger number of designs being explored in the former vs. the latter. The high values in the distance from the argmax plot (Fig. 2f) further confirm this by showing that the optimization is found not only by exploitation but also by exploration. For example, the exploration phase improves the quality factor as observed at iteration × cost ~ 800.

## Nanofabrication for centimeter-scale PnC resonators

The 3 cm optimum design is fabricated on high-stress silicon nitride ($Si_3N_4$), deposited by low-pressure chemical vapor deposition (LPCVD) on 2 mm silicon wafers. The fabrication starts by transferring the desired geometry on a thin positive tone resist (AR-P 6200) by a lithographic step. Typically, electron beam lithography or photolithography is used to pattern the masking layer. Electron beam lithography allows higher resolution and smaller feature sizes compared to the optical counterpart, but it is prone to stitching errors for structures exceeding the writing field. With each writing field extending for $100\,\mu m$–$1\,mm$, our centimeter-resonators require more than 30 fields, leading to noticeable stitching errors. We then implemented an overlap between adjacent writing fields of 100 nm and controlled the dose at specific locations (more details can be found in Supplementary Information Section 3). This resulted in an accurate transfer of the desired geometry, avoiding the presence of stitching errors. Despite the electron beam lithography's superior resolution, photolithography is preferable for fast and cost-effective manufacture at a large scale. With this in mind, we constrained the minimum feature size of the nanoresonators at 500 nm, compatible with ultraviolet photolithography.

Next, the pattern is transferred to the $Si_3N_4$ layer using an inductively coupled plasma (ICP) etching process ($CHF_3 + O_2$) at room temperature (Fig. 3a) before removing the masking layer (Fig. 3b).

The most critical part of the process is then suspending the high-aspect-ratio fragile structures over the substrate without causing any fracture, stiction, or collapse. Typically, this step is performed by liquid etchants such as KOH, which selectively removes the silicon substrate. However, turbulences and surface-tension forces can lead to collapse, destroying the suspended structures[61]. Those forces depend on the surface area of the nanoresonators and thus increase with the aspect-ratio, drastically reducing the fabrication yield for centimeter-scale nanoresonators. To overcome those limitations, stiction-free dry release can be employed[34,75], where the silicon substrate is isotropically removed by plasma etching. Fluorine-based ($SF_6$) dry etching at cryogenic temperature is particularly suited in view of its high selectivity against $Si_3N_4$, for which it does not require any mask or additional cleaning steps. Nevertheless, geometries with extreme high-aspect-ratio require a large opening ($>50\,\mu m$) from the substrate to avoid attraction due to charging effects, not achievable by $SF_6$ plasma etching alone.

To this end, we first directionally etch the silicon substrate by employing a thick positive tone photoresist (S1813) as a proactive layer (Fig. 3c). The step is carried out with cryogenic deep reactive ion etching (DRIE) using $SF_6 + O_2$ plasma[76]. Cryogenic DRIE allows the control of the opening size from the substrate without affecting the $Si_3N_4$ film quality. However, the photoresist is vulnerable to cracking in cryogenic DRIE[77]. To circumvent this limitation, the centimeter scale nanoresonators are shielded by an outer ring, which stops the cracks and prevents them from reaching the nanoresonators (see Supplementary Information Section 4 for details). The photoresist is then stripped off (Fig. 3d), and a hot piranha solution consisting of sulfuric acid and hydrogen peroxide is employed to remove residual contaminants on the surface. After that, a hydrofluoric acid solution allows the removal of oxides from the surface, which would otherwise prevent an even release of nanoresonators. Finally, the centimeter scale nanoresonators are suspended by a short 32 s fluorine-based ($SF_6$) dry etching step (Fig. 3e, f) performed at $-120\,°C$. The process isotropically etches the silicon substrate employing a pressure of 10 mbar, an ICP power of 2000 W, and a gas flow of 500 sccm, while the RF power is set to 0 W.

The developed process enables us to achieve a fabrication yield as high as 93% for our best chip, while the average value among all the fabricated devices is 75%. The main limiting factor is ensuring a particle-free surface prior to the $SF_6$.

## UHV lensed-fiber optical setup

To assess the mechanical properties of the fabricated nanoresonators, we characterized the strings' out-of-plane displacements by a laser interferometer (Fig. 4c). In it, 10% of the laser power of a 1550 nm infrared laser is focused on the nanoresonator. The reflected signal is collected by a lensed fiber to interfere with the local oscillator (LO) signal consisting of the remaining 90% of the infrared laser. The output signal, proportional to the resonator's displacement, is read out by an electronic spectrum analyzer after being converted by a balanced photodetector. The same output signal acts as an error function for the feedback loop employed to stabilize the phase of the setup against slow fluctuations caused by mechanical and thermal drift. The feedback loop is implemented with a PID controller, which adjusts the phase of the LO signal.

To avoid gas damping, increasingly dominant for high-aspect-ratio structures (see Supplementary Information Section 1 for details), the nanoresonator is placed inside an ultra-high vacuum chamber capable of reaching $P < 1 \times 10^{-9}$ mbar. The vacuum chamber is equipped with a 3-axis nanopositioner, which allows the alignment of the device with respect to the lensed fiber.

The chip is placed near a piezoelectric actuator which can vibrate out-of-plane. The latter allows the mechanically excite specific resonance frequencies of the nanoresonators. We then experimentally evaluate the quality factor of the nanoresonators by applying a sinusoidal function to the piezoelectric actuator at a frequency near the eigenfrequency of the localized mode. Once the displacement at resonance reaches a plateau, the excitation is abruptly turned off to measure the ringdown of the mechanical mode.

The power of the infrared laser can be manually adjusted by a variable optical attenuator, enabling to vary the laser power incident on the nanoresonators inside the vacuum chamber. This feature allows to perform ringdown measurements at different laser powers, which is crucial to rule out any optothermal or optical effects of the incident laser signal on the measured $Q$ factor (Supplementary Information Section 6).

## Data availability

The data supporting the findings of this study are available in a Zenodo database with the DOI identifier https://doi.org/10.5281/zenodo.10518818.

## Code availability

All code associated with this study is available from the corresponding authors, M.A.B. and R.A.N., upon request.

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

## Acknowledgements

A.C. and R.A.N. acknowledge valuable support from the Kavli Nanolab Delft, in particular from Charles de Boer and Roald van der Kolk. A.C. and R.A.N. would like to thank Matthijs H.J. de Jong and Minxing Xu for stimulating discussions and early assistance with fabrication and experiments. This work has received funding from the EMPIR program, co-financed by the Participating States, and from the European Union's Horizon 2020 research and innovation program (No. 17FUN05 Photo-Quant). This publication is part of the project, Probing the Physics of Exotic Superconductors with Microchip Casimir experiments (740.018.020) of the research program NWO Start-up, which is partly financed by the Dutch Research Council (NWO). Funded/Co-funded by the European Union (ERC, EARS, 101042855). Views and opinions expressed are, however, those of the author(s) only and do not necessarily reflect those of the European Union or the European Research Council. Neither the European Union nor the granting authority can be held responsible for them. D.S., M.A.B., and R.A.N. would like to acknowledge the TU Delft's 3mE Faculty Cohesion grant that enabled me to start this project.

## Author contributions

A.C., D.S., M.A.B., and R.A.N. designed the research; D.S., L.G., and M.A.B. conducted the data-driven computational design with support from A.C.; A.C. fabricated the nanomechanical resonators with support from R.A.N., A.C., and R.A.N led the experiment with support from D.S.; A.C. and D.S. analyzed the data; A.C., D.S., L.G., P.G.S., M.A.B., and R.A.N. wrote the paper.

## Competing interests

The authors declare no competing interests.
