## [Peer Review File · Nature Communications]

Centimeter-scale nanomechanical resonators with low dissipationREVIEWER COMMENTS

Reviewer #1 (Remarks to the Author):

In “Centimeter-scale nanomechanical resonators with low dissipation,” the authors report on an improved nanofabrication technique that allows engineering of strained Si₃N₄ thin film resonators with cross-sectional dimensions greater than 1 centimeter and thicknesses less than 100 nm. They use this technique to engineer a “soft-clamped” phononic crystal nanobeam with a defect length of several millimeters—similar in design to the beam reported in [30], but several times larger. Consistent with established-scaling laws for flexural modes of strained thin films (Eq. 1), the authors observe a ~7-fold increase in the mechanical Q factor of the defect mode over that reported in [30], to 6.6 billion. The authors claim that this is the highest room temperature Q factor on record for a suspended nanomechanical resonator. While only marginally so (see below), it is nevertheless intriguing for a variety of reasons.

With regards to suitability for Nature Communications, the reported results deserve attention because

1. Fabricating high-aspect ratio nanostructures under high tensile stress is notoriously challenging. The authors achieve an aspect ratio as high as $5e5$, which is on par with the best reported in the literature. For example, it is 40% larger than reported in [30] ($3.5e5$), and equivalent to that reported in [40].
2. The reported Q factor (6.6 billion) appears to be the highest in the literature for a room temperature suspended mechanical resonator. The nearest appears to be 3.6 billion, in [27], using a more compact “perimeter” mode design.
3. The numerical optimization technique used by the authors to design their resonator—though previously reported by the authors [29]—bears re-emphasis in this challenging context, and is a la mode.
4. The devices may find application in room temperature quantum experiments and in fundamental experiments in which the Q-mass factor is more important than the Q/mass factor (such as short-range gravity tests, spontaneous wavefunction collapse tests, and optomechanical dark matter searches.)

Unfortunately, I don’t think that the paper is suitable for publication, for three main reasons

5. To the extent that this is a yardstick paper (fabrication is not described until pg 4), it must be admitted that the advances are incremental. With regards to size: The cross-sectional size of the devices is greater by a factor of around 4 than similar strained thin film resonators (e.g. [30] and [57]); however, their aspect ratio is on par with devices made of thinner material (e.g. [40]). With regards to dissipation: The reported Q appears to be a record; however, only by a factor of 1.8 and only at room temperature [27]. By contrast, $Q > 10$ billion has been realized for strained [40] and unstrained [41] devices at cryogenic temperatures, and for levitated devices at room temperature (arxiv.org/abs/2304.02408).
6. Unless I’m mistaken, measurements for only a single device are reported in the main text and SI. While not unprecedented (see [41] or arxiv.org/abs/2304.02408), it is uncustomary for studies of nanomechanical dissipation to base evidence on a single device. ([41] took place in strenuous cryogenic conditions and arxiv.org/abs/2304.02408 was for a levitated device). Moreover, one wonders why only a single device was characterized, if the authors claim that their fabrication yield is greater than 75%.
7. While not unusual for young scientists to indulge in hyperbole—especially when submitting to high impact journals—I must admit I find the authors’ use excessive. There are sober reasons why the reported results deserve attention from the nanomechanics and optomechanics community. In their efforts to sell their results, however, the authors have embellished facts, and in many cases, made incorrect statements.

While I am unable to recommend publication in Nature Comm, I do think that the result is exciting.

In the spirit of peer review, below is a list of specific comments to help improve the manuscript

(page 1)

1. The authors' analogy to LIGO mirror suspensions, though apt, is imprecise. Please note that there are two types of resonators involved in a LIGO mirror vibration isolation systems---the string-like mirror suspension itself (which the authors refer to), and the pendulum formed by the mirror-suspension pair. Both experience dissipation dilution due to gravity (indirectly in the first case), and both play a role in LIGO's displacement sensitivity (suspension violin modes manifest as thermal noise peaks in the detection band, thermal motion of the mirror-suspension pendulum manifests as broadband thermal noise in the detection band). When the authors refer to the "landmark demonstrations of quantum effects at an unprecedented kilogram scale," they are referring to the pendulum, not the suspension resonator.
2. The authors should clarify what they mean by centimeter-scale, nano-scale, macroscale, etc.. Evidently 0.7 cm (common for nanobeams and trampoline) or 0.5 cm (commercially available from Norcada) is not centimeter-scale in their parlance. That's fine. Just say so. And what does it mean that "At the nanoscale, resonators up to millimeters long [...] are built on a chip"? Relatedly, in engineering literature, I think "macromechanics" (Fig. 1a) usually means something different than the authors mean.
3. The authors refer to Si₃N₄ as the "optimal" high stress material for nanomechanics at room temperature. It seems to me that it's a pragmatic choice. If it's "optimal," then the authors should specify a parameter space over which this optimization has been performed.
4. The authors claim that "nanomechanical resonators have been restricted to millimeter lengths, limiting their achievable Q factor and operational frequency". It's not clear to me what the authors mean by "the frequency is limited". If they mean "limited to high frequency", then one can achieve---and many have achieved--- lower frequencies by using low stress or no-stress (e.g. cantilever) geometries.

(page 2)

5. The authors present "multi-fidelity Bayesian optimization" as a pathway to improved resonator design at the centimeter scale. This is certainly interesting, but it's not a priori evident why increasing the size of the resonator requires numerical optimization *per se*. The resonator design in [30] (a 7 mm device similar to theirs) was semi-analytical (see [60]) and its measured Q factor closely matched theory on a wide variety of devices. The authors later claim, but do not provide evidence as best I can tell, that extrapolating these devices to larger dimensions results in (relatively) worse performance.
6. By "extreme high vacuum" (XHV, < 1e-11 mbar), the authors mean "ultra-high-vacuum" (UHV, <1e-9 mbar). This is one of many examples in the text where the authors use of hyperbole leads to misleading or incorrect statements.
7. Unless I'm mistaken, "aspect ratio", "clamping loss", and "bending loss" are not explicitly defined in the main text. The meaning is known to experts, but the article is currently written for non-experts.

(page 3)

8. The authors refer to Eq. 1 as a formula describing the quality factor of “strained thin one-dimensional mechanical resonators”. Presumably the phrase in quotes means “a string”? Consider simplifying or else add a citation describing what they mean.

9. Related to 8, note that Eq. 1 does not directly apply to phononic crystal (PnC) nanobeams of length L . In the case of a PnC nanobeam, a more appropriate (albeit approximate) definition of L is the length of the defect. This distinction is important because (by introducing Eq. 1) the authors are implicitly claiming to have made $L > 1$ cm resonators, when in fact---it may be argued---they have made $L < 1$ cm resonators embedded in a $L > 1$ cm PnC. Incidentally, this would be a good location for the authors to define what they mean by aspect ratio as L/h (which would force them to contend with the above point).

10. The authors should explain what they mean by “curvature” and at what “region” (the region near the clamps) where it is exaggerated. References here might be useful.

11. As mentioned above, it is unclear to me why “extreme-aspect-ratios [sic] and intricate cross-section variations necessitate high-fidelity computer simulations, rather than analytical models.” A central tenant of nanomechanical engineering is that continuum mechanics remains more or less well-behaved at the nanoscale, which is why finite element simulations work. It may very well be that there are analytically impractical considerations (such as von-Mises stresses) that become more important at the nanoscale, and that these drive the need for numerical simulations; however, I can think of no reason (and the authors provide none, as best I can tell) why the transition should occur at ~ 1 centimeter.

(page 4)

12. The authors state “Our method’s significant advantage is shown in Fig. 2d, as it finds designs with Q exceeding 10 billion---a first of its kind result” What is the first of its kind result here? A numerical simulation that predicts $Q > 10$ billion for a nanostring? Or do the authors mean that $Q > 10$ billion *would be* a novel result, were it realized in practice (at room temperature)?

13. The authors claim (if I understand correctly) that their optimization routine recovers the physically motivated design of [30], with the main exception being that the width of the defect is larger than in [30]. This is very interesting. They go on to claim that the thinner defect gives high Q but “worse performance.” How much worse (for those of us unable to play the Supplementary Video)?

14. The authors emphasize that their centimeter structures are enabled by an SF6 dry-release method plus a bunch of other clever hacks including a 70-um-thick “scaffolding” (the authors are the best in the business!). The details of the crucial SF6 dry-release are however not included in the manuscript or Methods. The authors cited three references [28,63,64], but none of them have the etch details. Perhaps they have a patent conflict? In any case, it would be nice to include details like temperature, pressure, gas flow rates, and etch timings.

(page 5)

15. In their discussion of Q factor measurements, the authors refer to devices (plural); however, they only present measurements for a single device. This is an important discrepancy and should be clarified.

16. Can the authors comment on where their measured $Q = 6.6$ billion compares to theory, and why they did not take other measurements to build statistics?

17. “highest value yet measured for a solid state resonator” should be qualified. The authors are presumably referring to room temperature, non-levitated solid state systems (arxiv.org/abs/2304.02408 is an example of $Q = 10^{10}$ for a room temperature levitated system)

18. It is well known that devices with the dimensions that the authors use are highly susceptible to photothermal heating, and that this heating can produce anti-damping forces which artificially increase or decrease the Q factor. Because the measured Q is not what was modeled, and especially since the authors only present a single measurement, they should comment on why photothermal heating was ruled out as possible artifact. The authors purport to provide evident in Fig. S8. Maybe it’s my pdf viewer, but I don’t see any data in that figure except a few points in the top left corner. Also that section of the appendix (F) contains several typos (e.g. “varied the laser power from 40 μm to 400 nm”?).

(page 6)

19. I’m not sure “quantum application” is a thing.

20. Allan Deviations must be defined relative to a specific variable and time. It is not clear what the Allan Deviation means here (frequency Allan Deviation? fractional frequency Allan Deviation? apparent force Allan Deviation?) or what the integration time is (1 sec, 10 sec?). Judging by the fact that it is unitless, I’m guessing a fractional frequency Allan Deviation of $1e-18$ at 1 second. However, that number would be better than an atomic clock...

(page 7)

21. It is indeed interesting that levitated nanoparticles require lower gas pressures to achieve $Q \sim 10^{10}$. But *why* is this, physically?

Reviewer #2 (Remarks to the Author):

The authors of 'Centimeter-scale nanomechanical resonators with low dissipation' demonstrate a new approach to design and fabricate mechanical resonators with very high aspect ratio. Using Bayesian optimization on two different, but correlated, length scales, they manage to reduce the simulation time required to find ideal geometries. By applying innovative fabrication methods, they are then able to experimentally realize these ideal geometries and obtain quality factors close to the numerical prediction.

The paper contains several novel concepts and the authors achieve record quality factors at room temperature. The paper is written in a clear way and contains all required information. I can find no technical error. In some cases, I personally found the language a bit close to hyperbole – as the results are clearly good, the use of words like 'myriad' does not seem necessary to sell the results. Nevertheless, I clearly recommend publication if the authors can address the following points:

1. There is a large body of literature to cite regarding atomic force microscopy. The citations [3,4] are not representative of this large community, and especially of the pioneering experiments from the late last century. For ultrasensitive AFM, I recommend to explicitly cite papers such as the nanowire scanning force microscope of the Poggio group (Rossi 2017) or similar experiments by the Budakian group (Nichol 2012).
2. 'To date, all centimeter scale mechanical resonators have been limited to micron-scale thicknesses.' Ghadimi et al. fabricated strings with a thickness of 20 nm and 7 mm length, which is only a factor 4 shorter than the devices in the present work. In my opinion, the sentence should be phrased more carefully.
3. The demonstrated devices are impressive, but the motivation of centimeter-scale resonators, as stated on page 1, is a bit vague. In the current formulation, it hinges on dark matter detection (a purely hypothetical idea) and room-temperature quantum mechanics (which on the fundamental side will probably not teach us anything new). Application in time-keeping will likely be limited by other factors than thermomechanical motion. For a journal like Nature Communications, the motivation in the introduction should be very clear.
4. 'The Q factor of the 3 cm design was a hundred times larger on average, but the optimal designs for both scales differed (Fig. 2b).' Looking at Fig. 2(b), I can see that the ratio of the quality factors is not uniform, but I cannot assess directly how the optimal designs differ. The sentence would in fact indicate that the scaling method between short and long designs does not work well. Can the authors comment on this apparent contradiction?
5. The central theme of the paper is the application of machine learning on two scales to find a design for a long, tapered beam. The result, as the authors write, is similar to the one of Ghadimi et al from 2018. The obvious question to be asked at this point is whether simply making the beam from the EPFL group longer by a factor 4 would have resulted in a similar quality factor $Q > 10^9$, or whether the algorithm provided a better value that human-driven design would have missed.
6. How realistic is the expected thermomechanical-limited frequency stability? Have the authors measured the frequency fluctuations at these comparably low resonance frequencies?

Responses to the comments of reviewers
Centimeter-scale nanomechanical resonators with low dissipation
reference number No. NCOMMS-23-40247

We thank the editor and reviewers for their enthusiasm for our results and for feedback, which has helped to improve the article significantly. Reviewers' comments are in bold, responses to the reviewers are in black, and added or modified sentences are in italic and magenta. To facilitate a clearer identification of each modification made to the manuscript, we have introduced line numbers throughout the document with the aim to assist reviewers in pinpointing and referencing specific changes easily.

Reviewers' comments:

Reviewer #1

In “Centimeter-scale nanomechanical resonators with low dissipation,” the authors report on an improved nanofabrication technique that allows engineering of strained Si₃N₄ thin film resonators with cross-sectional dimensions greater than 1 centimeter and thicknesses less than 100 nm. They use this technique to engineer a “so-clamped” phononic crystal nanobeam with a defect length of several millimeters—similar in design to the beam reported in [30], but several times larger. Consistent with established-scaling laws for flexural modes of strained thin films (Eq. 1), the authors observe a 7-fold increase in the mechanical Q factor of the defect mode over that reported in [30], to 6.6 billion. The authors claim that this is the highest room temperature Q factor on record for a suspended nanomechanical resonator. While only marginally so (see below), it is nevertheless intriguing for a variety of reasons. With regards to suitability for Nature Communications, the reported results deserve attention because:

a. Fabricating high-aspect ratio nanostructures under high tensile stress is notoriously challenging. The authors achieve an aspect ratio as high as $5e5$, which is on par with the best reported in the literature. For example, it is 40% larger than reported in [30] ($3.5e5$), and equivalent to that reported in [40].

b. The reported Q factor (6.6 billion) appears to be the highest in the literature for a room temperature suspended mechanical resonator. The nearest appears to be 3.6 billion, in [27], using a more compact “perimeter” mode design.

c. The numerical optimization technique used by the authors to design their resonator—though previously reported by the authors [29]—bears re-emphasis in this challenging context, and is a *la mode*.

d. The devices may find application in room temperature quantum experiments and in fundamental experiments in which the Q-mass factor is more important than the Q/mass factor (such as short-range gravity tests, spontaneous wavefunction collapse tests, and optomechanical dark mater searches.)

We thank the reviewer for the constructive feedback and review of our work. We have addressed the following comments, added measurements from additional fabricated devices and improved the manuscript based on the suggestions. The reviewer can find below our detailed responses and modifications to the manuscript.

Unfortunately, I don't think that the paper is suitable for publication, for three main reasons

We thank the reviewer for the careful assessment. We have revised our manuscript to address the three points raised by the reviewer to argue and explain its suitability for publication better.

e. To the extent that this is a yardstick paper (fabrication is not described until pg 4), it must

be admitted that the advances are incremental. With regards to size: The cross-sectional size of the devices is greater by a factor of around 4 than similar strained thin film resonators (e.g. [30] and [57]); however, their aspect ratio is on par with devices made of thinner material (e.g. [40]). With regards to dissipation: The reported Q appears to be a record; however, only by a factor of 1.8 and only at room temperature [27]. By contrast, $Q > 10$ billion has been realized for strained [40] and unstrained [41] devices at cryogenic temperatures, and for levitated devices at room temperature (arxiv.org/abs/2304.02408).

Thank you for your thorough evaluation. Our work aims to set out a route for reaching higher Q s in nanomechanical devices by scaling them up – a direction we have found to be unexplored. This creates challenges from a fabrication perspective, but also from a computational design perspective because larger devices with extreme aspect ratios lead to time-consuming simulations, impairing the data-driven optimization process. We believe that the strength of the work lies in providing solutions to both challenges.

The main innovation of the work is to show that the Q -factor of nanomechanical resonators can be significantly improved by increasing their lateral dimensions more than reducing the thickness. Equation 1 now illustrates this better, showing that the intrinsic (Q) of the material reduces with thinner resonators ($Q_{\text{int}} \propto h$).

$$Q = Q_{\text{int}} \left[n^2 \pi^2 \frac{E}{12\sigma} \left(\frac{h}{L} \right)^2 + 2 \sqrt{\frac{E}{12\sigma}} \left(\frac{h}{L} \right) \right]^{-1} \propto \left[n^2 \pi^2 \sqrt{\frac{E}{48\sigma}} \left(\frac{h}{L^2} \right) + \frac{1}{L} \right]^{-1} \quad (1)$$

We argue that the approach of further increasing device size to increase Q is not an approach that is generally adopted at the moment, and highlighting and demonstrating its potential for record Q is therefore valuable to the community. We believe that this is also not a trivial route, since achieving a fourfold increase in length represents a significant advance in nano-fabrication and design, especially in a competitive field that has struggled to surpass the centimeter scale for at least half of a decade. This is shown in the following plot, representing the evolution of the length of SiN nanomechanical resonators possessing high Q over time:

The advance we provide to the field lies in the ability to extend these devices to multi-centimeter lengths while retaining good yield, a feat previously thought not practically feasible from a nano-manufacturing standpoint. In principle, our method can be extended to longer scales, but we strategically chose 3 cm since going beyond this length for lower dissipations would require lower vacuum levels, and significantly more infrastructure than our turbo (and roughing) pump. We agree with the suggestion of the reviewer that the nanofabrication challenge should get more prominence and have therefore highlighted it earlier on in the manuscript.

In terms of comparison to the performance in prior work, we are happy to have set a new record but we are hopeful that further work along the presented length scaling route might even lead to higher records by others. Furthermore, as pointed out by the reviewer, our approach relieves the temperature and vacuum requirements to reach these high- Q factors, unlike other investigations. We demonstrate that record-low dissipation levels can be reached at room temperature, previously accessible only in environments demanding costly liquid-helium infrastructure. Previous references [40] and [41] (Nat. Phy. 13, 2022 and Nat. Com. 11(3373), 2020) operate at cryogenic temperatures, whose operation cannot be compared to the reduced cost and complexity of room temperature operation. This is particularly noteworthy considering that the quality factor for materials like

silicon nitride have been consistently reported to increase at cryogenic temperatures ^{1,2}, as now mentioned in the conclusion of the manuscript. In the case of levitated particles at room temperature, these systems require vacuums 100 times lower than ours and lowering a vacuum from 10^{-9} mbar to 10^{-11} mbar requires significantly more complex infrastructure. Levitated nanospheres can only achieve quality factors of 10 billion with some of the lowest vacuums maintained on Earth. This requires specialized chambers, pumping, and baking infrastructure that is very different from the relative ease of reaching 10^{-9} mbar. Our on-chip platforms offer state-of-the-art performance in a regime that requires significantly less complex infrastructure.

Given the reviewer’s comments, the manuscript has been rewritten to better articulate these points, emphasizing our contribution to the field in terms of both technological advancement and practical application.

f. Unless I’m mistaken, measurements for only a single device are reported in the main text and SI. While not unprecedented (see [41] or arxiv.org/abs/2304.02408), it is uncustomary for studies of nanomechanical dissipation to base evidence on a single device. ([41] took place in strenuous cryogenic conditions and arxiv.org/abs/2304.02408 was for a levitated device). Moreover, one wonders why only a single device was characterized, if the authors claim that their fabrication yield is greater than 75%.

Thank you for bringing this to our attention. Although our fabrication yield exceeds 75%, our measurement routine limits the number of devices we can test. This is due to the lengthy ringdown time for each device and the difficulty in maintaining a stable measurement over hours to get clean ringdown measurements.

Following your comment, we have added a new section J (see below) to the Supplementary Information to show measurements for 3 devices with multiple measurements each, to show their consistency. The devices have the same optimized geometry described in the main text and have been fabricated with the same fabrication steps detailed in both the main text and the methods section. All these devices exhibit quality factors exceeding 4.6 billion at room temperature, with a variation shown in a new Fig. S12, consistent with previous studies on silicon nitride (see for example ^{3,4,5,6}).

(New section: Supplementary Information J) *Quality factors of additional devices*

This section presents the measurements conducted on additional fabricated devices, which share the same optimized design and are lithographically identical to the device featured in Fig. 4 of the main text. Figure S12 shows multiple ringdown traces from 3 devices. The uppermost blue curves correspond to four different measurements of the device discussed in the main text, while the green and light purple curves depict measurements obtained from two additional devices, each measured twice. Traces for the same device exhibit a comparable decay rate, resulting in the same quality factor, which is indicated above the first curve of each device. Linear fits for each trace are depicted as solid black lines. The power values on the y-axis are adjusted relative to the maximum value of each curve before applying a 10 dBm offset between them. An additional 10 dBm spacing along the y-axis distinguishes different devices. Strong fluctuations resulting from unwanted temperature drifts and mechanical vibrations of the setup can push the measured signal outside the linear region of the interference signal for a brief time interval. This, in turn, leads to occasional spikes, as visible in some curves.

The results demonstrate a consistent and reproducible quality factor among different measurements of each device. The observed variation among the quality factor of different devices suggests differences in surface quality among the devices, e.g., surface impurities and surface roughness ²¹, or different thicknesses. As further detailed in Sec. B, variation in the final thickness of each device might occur due to the difficulties in dissipating heat during the undercut process, significantly impacting the measured quality factor. Among all the devices fabricated and measured, we observed a variation in quality factor of around one order of magnitude, which is in line with typical variations found in other works.

¹Thomas Gisler, Mohamed Helal, Deividas Sabonis, Urs Grob, Martin Héritier, Christian L. Degen, Amir H. Ghadimi, and Alexander Eichler, Phys. Rev. Lett. 129, 104301

²Yuan, Mingyun, Martijn A. Cohen, and Gary A. Steele. Applied Physics Letters 107.26 (2015)

³Bereyhi, M.J., Beccari, A., Groth, R. et al. Hierarchical tensile structures with ultralow mechanical dissipation. Nat Commun 13, 3097 (2022).

⁴Høj, D., Wang, F., Gao, W. et al. Ultra-coherent nanomechanical resonators based on inverse design. Nat Commun 12, 5766 (2021)

⁵Tsaturyan, Y., Barg, A., Polzik, E. et al. Ultracoherent nanomechanical resonators via soft clamping and dissipation dilution. Nature Nanotech 12, 776–783 (2017).

⁶A. H. Ghadimi et al., Elastic strain engineering for ultralow mechanical dissipation. Science360,764-768(2018)

Figure S12: Multiple ringdown traces of 3 fabricated nanoresonators with the optimized design presented in the main text. Traces corresponding to measurements from the same device are color-coded for easy identification.

To facilitate comparison, each trace has been vertically shifted relative to its respective maximum value, $(P(t) - P_{max})$. Subsequently, an incremental 10 dBm offset has been applied to each trace, starting with the top-most blue trace. An additional 10 dBm offset is applied between measurements from different devices.

g. While not unusual for young scientists to indulge in hyperbole—especially when submitting to high impact journals—I must admit I find the authors’ use excessive. There are sober reasons why the reported results deserve attention from the nanomechanics and optomechanics community. In their efforts to sell their results, however, the authors have embellished facts, and in many cases, made incorrect statements.

We sincerely apologize for the hyperbolic statements. We now recognize that our enthusiasm could easily be misinterpreted by lack of objectivity. Thank you for your candid feedback. We have carefully reviewed the manuscript to remove hyperbolic statements and enhance the clarity and rigor of the language. We have revised and corrected sentences that could cause confusion or convey incorrect statements. The reviewer can find all the updates and overall revisions in the main manuscript based on the detailed comments suggested for the peer review.

While I am unable to recommend publication in Nature Comm, I do think that the result is exciting. In the spirit of peer review, below is a list of specific comments to help improve the manuscript.

We appreciate the reviewer for the detailed comments on our work which we feel have improved the paper. We have addressed all the comments carefully to improve the manuscript. The reviewer can find below our detailed responses and modifications to the manuscript.

(page 1)

1. The authors’ analogy to LIGO mirror suspensions, though apt, is imprecise. Please note that there are two types of resonators involved in a LIGO mirror vibration isolation systems—the string-like mirror suspension itself (which the authors refer to), and the pendulum formed by the mirror-suspension pair. Both experience dissipation dilution due to gravity (indirectly in the first case), and both play a role in LIGO’s displacement sensitivity (suspension violin modes manifest as thermal noise peaks in the detection band, thermal moon of the mirror-suspension pendulum manifests as broadband thermal noise in the detection band). When the authors refer to the “landmark demonstrations of quantum effects at an unprecedented kilogram scale,” they are referring to the pendulum, not the suspension resonator.

Thank you for your comment. Originally, we aimed to create an analogy to the mirror-suspension pair resonator utilized in LIGO and Advanced LIGO rather than the string-like resonator alone. The reported quality factor for Advanced LIGO resonators of 10^8 refers to the pendulum-like motion of the resonators at 0.43 Hz. Following your feedback, we have revised the introduction to provide a more precise description of the resonators we are discussing, as outlined below:

(line 43-44) ... *pendulum formed by the mirror-suspension pair* ...

(line 49-50) *The resulting high quality factor of the pendulum modes of the order of 10^8 allows isolating the detector from thermomechanical noise, enabling it to reach its enhanced displacement sensitivity.*

(line 54-55) *...contribute to the first observations of gravitational waves.*

2. The authors should clarify what they mean by centimeter-scale, nano-scale, macroscale, etc.. Evidently 0.7 cm (common for nanobeams and trampoline) or 0.5 cm (commercially available from Norcada) is not centimeter-scale in their parlance. That’s fine. Just say so. And what does it mean that “At the nanoscale, resonators up to millimeters long [...] are built on a chip”? Relatedly, in engineering literature, I think “macromechanics” (Fig. 1a) usually means something different than the authors mean.

Thank you. We have updated the main text to provide more precise definitions for the terms used. In Fig. 1, the term ‘macromechanics’ is now referred to as ‘macroscale resonators,’ and ‘nanomechanics’ has been similarly adjusted for clarity. Furthermore, we have provided more precise definitions for these terms in the introductory paragraph and included a range value in the caption of Fig. 1. The term ‘centimeter-scale’ refers to lateral dimensions larger than 1 cm. We have clarified this definition for better understanding and comparison with previous devices. Lastly, the sentence “At the nanoscale, resonators up to millimeters long [...] are built on a chip” refers to the possibility of integrating nanomechanical resonators on a chip, contrary to macroscale resonators, which are generally free-standing cables or strings. We have revised the text to clarify the meaning of the sentence.

(line 40-42) *... resonators with macroscopic lengths on the order of centimeters and above^{2,19} and resonators with nanometers thicknesses^{20–22}.*

(line 58-60) *To date, resonators in this category, which span lengths of several centimeters, have been limited to micrometers minimum thicknesses.*

(line 64-66) *At the nanoscale, resonators possess significantly reduced thickness, in the range of nanometers, with a length limited to millimeters.*

Figure 1

3. The authors refer to Si3N4 as the “optimal” high stress material for nanomechanics at room temperature. It seems to me that it’s a pragmatic choice. If it’s “optimal,” then the authors should specify a parameter space over which this optimization has been performed.

Thank you for your comment, we agree. We have removed the term ‘optimal’ and expanded in the main text on the advantages of using silicon nitride as detailed below:

(line 73-76) *...one of the most common and easily manufacturable materials in view of its advantageous optical and mechanical properties at room temperature.*

4. The authors claim that “nanomechanical resonators have been restricted to millimeter lengths, limiting their achievable Q factor and operational frequency”. It’s not clear to me what the authors mean by “the frequency is limited”. If they mean “limited to high frequency”, then one can achieve —and many have achieved— lower frequencies by using low stress or no-stress (e.g. cantilever) geometries.

We understand that this statement requires additional clarification, thank you. We edited it accordingly:

(line 85-87) *Generally, the quality factor in this class of resonators (Fig. 1d) is proportional to the length, and the resonance frequency is inversely proportional to the length*^{21,31}.

(page 2)

5. The authors present “multi-fidelity Bayesian optimization” as a pathway to improved resonator design at the centimeter scale. This is certainly interesting, but it’s not a priori evident why increasing the size of the resonator requires numerical optimization per se. The resonator design in [30] (a 7 mm device similar to theirs) was semi-analytical (see [60]) and its measured Q factor closely matched theory on a wide variety of devices. The authors later claim, but do not provide evidence as best I can tell, that extrapolating these devices to larger dimensions results in (relatively) worse performance.

We appreciate the comment on the discussion about using numerical simulations to design the resonator. We detailed the discussion and updated the manuscript considering this comment as well as the comments in Response 11 of reviewer 1. As the review points are continuous, we detail this response in comment 11.

6. By “extreme high vacuum” (XHV, $< 1e-11$ mbar), the authors mean “ultra-high-vacuum” (UHV, $< 1e-9$ mbar). This is one of many examples in the text where the authors use of hyperbole leads to misleading or incorrect statements.

We thank the reviewer for pointing out the terminology regarding vacuum levels. We agree and have replaced extreme high vacuum as follow.

(line 140-141) *... require significantly more stringent vacuum conditions ...*

(line 460) *... vacuums levels as low as as 10^{-11} mbar.*

7. Unless I’m mistaken, “aspect ratio”, “clamping loss”, and “bending loss” are not explicitly defined in the main text. The meaning is known to experts, but the article is currently written for non-experts.

Based on the reviewer’s comment, we have added the mentioned definitions. The response is also with the comments for Response 10 for reviewer 1’s review.

(line 163-166) *For given aspect ratio h/L , the first and second term in the denominator of Equation 1 originates from bending loss and clamping loss, respectively, for which we use the definitions from Schmid et al.*²¹.

(line 170-175) *The clamping region is defined as the part of the resonator at a distance of less than $L_c = \sqrt{E/6\sigma}h$ from the clamping points where the mode shape deviates strongly from a simple sine function*²¹. *The curvature is defined as the reciprocal of the radius of curvature.*

(page 3)

8. The authors refer to Eq. 1 as a formula describing the quality factor of “strained thin one-dimensional mechanical resonators”. Presumably the phrase in quotes means “a string”? Consider simplifying or else add a citation describing what they mean.

As the reviewer stated, it corresponds to a string resonator. We updated this.

(line 155-156) *The quality factor of string resonators with a constant cross-section is given by*²¹

9. Related to 8, note that Eq. 1 does not directly apply to phononic crystal (PnC) nanobeams of length L . In the case of a PnC nanobeam, a more appropriate (albeit approximate) definition of L is the length of the defect. This distinction is important because (by introducing Eq. 1) the authors are implicitly claiming to have made $L > 1$ cm resonators, when in fact—it may be argued—they have made $L < 1$ cm resonators embedded in a $L > 1$ cm PnC. Incidentally, this would be a good location for the authors to define what they mean by aspect ratio as L/h (which would force them to contend with the above point).

We appreciate the need to carefully pointing out the discussion about equation 1. As the reviewer correctly mentions, Eq. 1 is designed for straight beam resonators and cannot be directly applied to phononic crystal (PnC) beams. However, it serves as an approximation that adequately captures the trends exhibited by PnC beams. The PnC beam will behave similarly to Eq. 1: its defect mode will resemble a higher mode number $n > 1$ in Eq. 1, where L is the total length of the phononic crystal just like in the experiment. We therefore want to highlight that we do not agree that we make an implicit claim that the size of the defect mode is larger than 1 cm. In order to make this clear, we have now provided more details to clarify that the defect mode we measure is not similar to the fundamental mode of a string resonator with length L and compare our resonator with other resonators that consider the fundamental mode or much smaller n for the resonance mode in the manuscript.

(line 182-190) *Those include phononic crystal-based string and membrane resonators, which employ a higher-order eigenmode confined in a central defect by the acoustic bandgap^{32,33}, spiderweb and perimeters resonators, which exploit low-order eigenmodes^{30,31}, or other methods considering the fundamental mode^{29,35} (Fig. 1a). It is important to notice though that Equation 1. is a straight beam formulation that does not directly apply to phononic crystal strings⁴⁹ but can give insight into general trends with h/L .*

10. The authors should explain what they mean by “curvature” and at what “region” (the region near the clamps) where it is exaggerated. References here might be useful.

Thank you. We detailed the explanation of each meaning. We updated this also when addressing Point 7 of reviewer 1.

11. As mentioned above, it is unclear to me why “extreme-aspect-ratios [sic] and intricate cross-section variations necessitate high-fidelity computer simulations, rather than analytical models.” A central tenant of nanomechanical engineering is that continuum mechanics remains more or less well-behaved at the nanoscale, which is why finite element simulations work. It may very well be that there are analytically impractical considerations (such as von-Mises stresses) that become more important at the nanoscale, and that these drive the need for numerical simulations; however, I can think of no reason (and the authors provide none, as best I can tell) why the transition should occur at ~ 1 centimeter.

Thank you for addressing the discussion regarding the increasing costs in numerical analysis as the aspect ratio rises throughout the design phase. To address this important concern, we have expanded the explanation along with the comments for Point 5 of reviewer 1.

We agree with the reviewer’s comment and acknowledge that the initial wording may have led to confusion. We do not assert that analytical models are unsuitable for high-aspect-ratio optimization but rather that designing the high-aspect ratio resonator with direct numerical simulation requires more expensive simulations than the low-aspect ratio resonator. Although a semi-analytical model could suffice for our specific 1D resonators due to their straightforward geometry, our focus is on providing a general method for designing and fabricating centimeter-scale resonators, adaptable to various designs beyond 1D phononic crystal strings. For example, analytical methods fall short when the geometrical complexity of the resonators increases in 2D structures because analytically calculating stress distribution and elastic energy becomes non-trivial, requiring numerical simulations. We have clarified this argument.

We clarified that designing high aspect ratio resonators with direct numerical simulation demands advanced design strategies due to the elevated computational cost of accurately capturing their behavior. Finally, we answer whether directly extrapolating the previous optimized design on a 7 mm resonator [33] could work following question 5 of Reviewer 1. During the revision (along with the response to the fourth point of Reviewer 2), we updated the description of Fig. 2b in more detail.

(line 199-210) *However, the resulting higher aspect-ratios demand advanced design strategies due to the*

elevated computational cost of accurately capturing their behavior through direct simulation models. Numerical analyses require increased degrees of freedom (DOFs) to describe models with extreme aspect ratios. For instance, conducting finite element analysis on a centimeter-long string with phononic crystals requires roughly ten times more DOFs than the same geometry with a one order lower aspect-ratio. This poses a challenge for centimeter-scale designs, as the heightened computational cost sets a practical boundary, particularly given the limited availability of high-fidelity data.

(line 217-224) *We utilize finite element simulations with COMSOL⁵³ to maximize the quality factor. Despite the existence of analytical derivations for one-dimensional beams⁴⁹, we discuss direct numerical simulations in this study. This choice aims to provide a more generalized design approach for large aspect-ratio resonators. The approach in this study could be expanded for other high Q resonators sensitive to the length scale but lacking derived analytical formulations^{30–32}.*

(line 244-255) *... confirming the expected correlation between the two models according to the soft clamped Q factor in Equation 1. The variation in Figure 2b underscores that what improves performance for a 3 mm resonator does not necessarily translate into a better 3 cm resonator. Therefore, exclusively relying on the low fidelity model (3 mm resonator) would not yield the desired outcome for designing the 3 cm resonator. Simultaneously, despite each resonator size having its unique set of optimal geometric parameters, the algorithm is still capable of learning enough from the response of the 3 mm resonator to guide the optimization of the 3 cm one.*

(page 4)

12. The authors state “Our method’s significant advantage is shown in Fig. 2d, as it finds designs with Q exceeding 10 billion—a first of it’s kind result” What is the first of its kind result here? A numerical simulation that predicts $Q > 10$ billion for a nanostring? Or do the authors mean that $Q > 10$ billion would be a novel result, were it realized in practice (at room temperature)?

We appreciate the reviewer for the proper comment. Our method corresponds to the MFBO, and the advantage corresponds to the fact that MFBO results outperform the regular BO results. We have now specified it in the manuscript more carefully, avoiding misleading descriptions.

(line 263-265) *The advantage of using multi-fidelity transformed Bayesian optimization is shown in Fig. 2d, as it finds a design with $Q \dots$*

13. The authors claim (if I understand correctly) that their optimization routine recovers the physically motivated design of [30], with the main exception being that the width of the defect is larger than in [30]. This is very interesting. They go on to claim that the thinner defect gives high Q but “worse performance.” How much worse (for those of us unable to play the Supplementary Video)?

Thank you for your insightful feedback. In the revised manuscript, we added the description of the optimized design’s vibration mode to answer the comment. The optimal design recovers the general concept of the physically motivated design of [33], having the tapering shape toward the center of the resonator. One important difference in our optimized design was discovering the optimal vibration mode in anti-symmetric rather than symmetric mode. Because the quantitative comparison between the symmetric optimized mode is infeasible with the absence of the optimized symmetric mode, we added a more detailed discussion elucidating the physics of having a wider defect width in the anti-symmetric optimized mode and clearly stated the optimum design. We also want to clarify that we mentioned worse performance while comparing not with thin defects, but with narrow clamping region. To avoid possible confusions, we removed the comparison and specified our claim with a more detailed explanation as follows.

(line 271-281) *The optimal design maximized the **unit cell** width around the clamping region and narrowed down the width of the **unit cells** coming near the center of the resonator. **One distinguishing aspect of our approach was that during the optimization under the design domain, the algorithm considered both symmetric and anti-symmetric modes, but the optimized mode was selected to be anti-symmetric rather than symmetric. Because of this difference, the defect width was not minimized, unlike the early study³³, since the kinetic energy with significant movement is not maximized at the center.***

14. The authors emphasize that their centimeter structures are enabled by an SF6 dry-release method plus a bunch of other clever hacks including a 70-um-thick “scaffolding” (the authors are

the best in the business!). The details of the crucial SF6 dry-release are however not included in the manuscript or Methods. The authors cited three references [28,63,64], but none of them have the etch details. Perhaps they have a patent conflict? In any case, it would be nice to include details like temperature, pressure, gas flow rates, and etch timings.

We appreciate the reviewer's comment, as it contributes to enhancing the reproducibility of our work. The temperature of the process is reported in reference [28], together with information on the high pressure and the low DC bias required to achieve high isotropic etching. We have now expanded the 'Method' section to provide additional information on the developed SF6 dry-release process. At the same time we have moved details concerning the "micro-scaffolding" and the lithographic steps from the "Method" section to the main text

(line 643-648) *Finally, the centimeter scale nanoresonators are suspended by a short 32 seconds fluorine-based (SF6) dry etching step (Fig. 3e-f) performed at -120°C . The process isotropically etches the silicon substrate employing a pressure of 10 mbar, an ICP power of 2000 W and a gas flow of 500 sccm, while the RF power is set to 0 W.*

(line 320-323) *This is achieved by employing multiple exposures with varying resolutions to reduce the exposure time without compromising the accuracy (Supplementary Information Sec. C).*

(line 348-350) *The micro-scaffolding is lithographically defined by a second exposure and transferred into the silicon substrate via deep directional plasma etch (Fig. 3c)*

(page 5)

15. In their discussion of Q factor measurements, the authors refer to devices (plural); however, they only present measurements for a single device. This is an important discrepancy and should be clarified.

Thank you for pointing it out. We have included measurements of additional devices in the Supplementary Information. To clarify that the ringdown and spectrum displayed in Fig. 4 were acquired from a single device, we added the following sentences to the "Low dissipation at room temperature" section:

(line 377) *... for the device depicted in Fig. 4a.*

(line 392) *... for our best performing device.*

16. Can the authors comment on where their measured $Q = 6.6$ billion compares to theory, and why they did not take other measurements to build statistics?

Thank you for your question. The expected quality factor from theory (eq. 1) and Finite Element Analysis for a 50 nm-thick resonator is 10 billion. We attribute the difference between the experimental and predicted value primarily to thicknesses variation as detailed in Section B of the Supplementary Information. An expected thickness variation of up to 20 nm aligns with the employed SF6 dry etching technique due to the difficulties in heat dissipation during the process. We updated the manuscript to explain more clearly the thickness difference between the simulation and experiment, and the resulting difference in the Q factor as detailed below.

In regard to additional devices, we added a new Section J. to the Supplementary Information with the measurements from additional devices. More details can be found on our reply to point f of reviewer 1.

(line 350-355) *While our simulated strings are designed with a nominal thickness of 50 nm, the fabricated resonators have a larger final thickness from edge to center along their length. This is caused by the difficulty of dissipating heat during the etching step of the silicon substrate (Supplementary Information Sec. B)*

(line 397-401) *To corroborate these findings, we tested additional lithographically identical devices, exhibiting the same frequency response in good agreements with the simulation, and quality factors within a 50 % range (Supplementary Information Sec. J)*

17. "highest value yet measured for a solid state resonator" should be qualified. The authors are presumably referring to room temperature, non-levitated solid state systems (arxiv.org/abs/2304.02408 is an example of $Q = 10^{10}$ for a room temperature levitated system)

We have substituted 'solid state resonator' with 'clamped resonators' as detailed below. In the updated version of the manuscript, this specific sentence has been moved to the conclusion section.

(line 423-425) ... *we experimentally achieve the lowest dissipation yet measured for clamped resonators at room temperature.*

18. It is well known that devices with the dimensions that the authors use are highly susceptible to photothermal heating, and that this heating can produce an-damping forces which artificially increase or decrease the Q factor. Because the measured Q is not what was modeled, and especially since the authors only present a single measurement, they should comment on why photothermal heating was ruled out as possible artifact. The authors purport to provide evident in Fig. S8. Maybe it's my pdf viewer, but I don't see any data in that figure except a few points in the top left corner. Also that second of the appendix (F) contains several typos (e.g. "varied the laser power from 40 um to 400 nm"?).

Thank you for your feedback. In Figure S8, the plot illustrates the comparable decay rates observed in ringdown measurements conducted with different laser powers on the optimized device. Specifically, the graph displays traces acquired with laser powers of 40 μW and 400 nW, both of which exhibit similar slopes. This observation suggests that photothermal effects do not significantly influence the measurements. We also want to emphasize that the experiment employs a laser operating at 1550 nm, where the absorption of LPCVD Silicon Nitride is minimal, and the laser power coupled to the resonator is as low as 400 μW . We have re-exported Figure S8 in bitmap instead of vector format to reduce the loading time in the pdf and included a copy of the figure below for your convenience. Additionally, we have corrected the typographical errors in Appendix F.

Figure S8

(page 6)

19. I'm note sure "quantum application" is a thing.

Thank you. We have removed the word 'quantum' from the sentence.

(line 433-434) *A natural application is ground state cooling of the mechanical resonators in a room-temperature environment.*

20. Allan Deviations must be defined relative to a specific variable and me. It is not clear what the Allan Deviation means here (frequency Allan Deviation? fractional frequency Allan Deviation? apparent force Allan Deviation?) or what the integration me is (1 sec, 10 sec?). Judging by the fact that it is unitless, I'm guessing a fractional frequency Allan Deviation of $1e-18$ at 1 second. However, that number would be better than an atomic clock...

We appreciate the reviewer's comment to add details about the Allan deviation. First, we mistakenly referred to Allan deviation in the manuscript while calculating the Allan variance. The updated version of the manuscript contains the correct calculation for the nondimensional Allan deviation⁷:

⁷Tomás Manzaneeque, Murali K. Ghatkesar, Farbod Alijani, Minxing Xu, Richard A. Norte, and Peter G. Steeneken Phys. Rev. Applied 19, 054074 – Published 23 May 2023

$$\sigma_y(\tau) = \sqrt{\frac{m_{\text{eff}} 2\pi f_0 k_B T}{A_0^2 Q^3} \cdot \frac{1}{\tau}} \quad (2)$$

Following the reviewer's comment, we have updated the conclusion of the manuscript as below:

(line 448-451) *Conservatively assuming sub-micrometers amplitude displacements in the linear regime, we can extrapolate a thermomechanical limited Allan deviation⁵⁸ $\sigma_y(\tau) \sim 3 \times 10^{-12} / \sqrt{\tau}$ for $m_{\text{eff}} = 4.96 \times 10^{-13}$ kg.*

(page 7)

21. It is indeed interesting that levitated nanoparticles require lower gas pressures to achieve $Q \sim 10^{10}$. But why is this, physically?

In our clamped nanomechanical resonators, the primary sources of dissipation are intrinsic effects, named 'bending losses' and 'clamping losses' in our manuscript in equation 1, while gas damping becomes negligible at pressures $\sim 1 \times 10^{-9}$ mbar within our frequency range. On the contrary, in the case of levitated objects, dissipation is typically limited by the collision rate between the particle and background gas molecules. Consequently, significantly lower pressures are required to achieve a quality factor (Q) on the order of 10^{10} . It's important to note that the operating frequency for levitated particles with high Q, as discussed in the work by Dania, Lorenzo, et al.⁸, differs from the frequency range of our devices, leading to differences in the role of gas damping. We have updated the manuscript as detailed below:

(line 138-143) *This solid-state platform performs on par with counterparts such as optically levitated nanospheres which require significantly more stringent vacuum conditions as low as 10^{-11} mbar⁴⁶ to reduce the collision rate between the particle and background gas molecules, typically the dominant source of dissipation.*

(line 456-462) *This comparison becomes even more striking when considering that levitated particles are essentially isolated from the environment, interacting only minimally with residual gas molecules at vacuum levels as low as 10^{-11} mbar, which requires complex infrastructure far beyond the turbo and roughing pump combination used for achieving 10^{-9} mbar. In stark contrast...*

⁸Dania, Lorenzo, et al. "Ultra-high quality factor of a levitated nanomechanical oscillator." arXiv preprint arXiv:2304.02408 (2023)

Reviewer #2

The authors of ‘Centimeter-scale nanomechanical resonators with low dissipation’ demonstrate a new approach to design and fabricate mechanical resonators with very high aspect ratio. Using Bayesian optimization on two different, but correlated, length scales, they manage to reduce the simulation time required to find ideal geometries. By applying innovative fabrication methods, they are then able to experimentally realize these ideal geometries and obtain quality factors close to the numerical prediction. The paper contains several novel concepts and the authors achieve record quality factors at room temperature. The paper is written in a clear way and contains all required information. I can find no technical error. In some cases, I personally found the language a bit close to hyperbole – as the results are clearly good, the use of words like ‘myriad’ does not seem necessary to sell the results. Nevertheless, I clearly recommend publication if the authors can address the following points:

We appreciate the reviewer for the detailed comments on our work. We have addressed all the comments carefully to improve the manuscript. The reviewer can find below our detailed responses and modifications to the manuscript.

1. There is a large body of literature to cite regarding atomic force microscopy. The citations [3,4] are not representative of this large community, and especially of the pioneering experiments from the late last century. For ultrasensitive AFM, I recommend to explicitly cite papers such as the nanowire scanning force microscope of the Poggio group (Rossi 2017) or similar experiments by the Budakian group (Nichol 2012).

Thank you for bringing the work of Rossi et al. and Nichol et. al. to our attention. We have included the suggested citations, along with supplementary references related to AFM and MRFM.

[3] Sidles, J. A. et al. *Magnetic resonance force microscopy. Reviews of Modern Physics* 67, 249–265 (1995).

[4] Binnig, G., Quate, C. F. & Gerber, Ch. *Atomic Force Microscope. Physical Review Letters* 56, 930–933 (1986).

[5] Rossi, N. et al. *Vectorial scanning force microscopy using a nanowire sensor. Nature Nanotechnology* 12, 150–155 (2017).

[6] Nichol, J. M., Hemesath, E. R., Lauhon, L. J. & Budakian, R. *Nanomechanical detection of nuclear magnetic resonance using a silicon nanowire oscillator. Physical Review B* 85, 054414 (2012).

2. ‘To date, all centimeter scale mechanical resonators have been limited to micron-scale thicknesses.’ Ghadimi et al. fabricated strings with a thickness of 20 nm and 7 mm length, which is only a factor 4 shorter than the devices in the present work. In my opinion, the sentence should be phrased more carefully.

Thank you. We intended to emphasize the significance and challenges associated with reaching the centimeter scale (above 1 cm) for nanomechanical resonators. While it is true that resonators with lengths approximately four times smaller have been developed, we consider 1 centimeter as a threshold for a different class of resonators and want to acknowledge the difficulties in achieving this particular scale. We understand that the sentence might appear misleading and we rephrased it more carefully to take your comment into consideration. At the same time, we have better defined the meaning of ‘centimeter-scale resonators’.

(line 58-63) *To date, resonators in this category, which span lengths of several centimeters, have been limited to minimum thicknesses on the order of micrometers. These types of mechanical resonators have been particularly successful in demonstrating low dissipation in room-temperature environments.*

3. The demonstrated devices are impressive, but the motivation of centimeter-scale resonators, as stated on page 1, is a bit vague. In the current formulation, it hinges on dark matter detection (a purely hypothetical idea) and room-temperature quantum mechanics (which on the fundamental side will probably not teach us anything new). Application in time-keeping will likely be limited by other factors than thermomechanical motion. For a journal like Nature Communications, the motivation in the introduction should be very clear.

Thank you for the suggestion. We updated the introductory paragraph to strengthen the motivation of our work. You can find the updated sentence below.

(line 87-97) *As nanoscale resonators become longer, their manufacturing yield drops dramatically due to compounding factors including limits in devices per chip, the alignment of nanoscale features over centimeters, and as resonators become longer they become increasingly delicate suspended structures. These combined decrease device yield and experimental match with expected Qs as resonators become more susceptible to small forces during nanofabrication. These overlapping fabrication challenges have made reliably increasing the length of nanoscale resonators beyond millimeter scales prohibitively difficult in practice.*

(line 98-118) *Advancing these resonators to multi-centimeter lengths while maintaining their nanoscale thickness would uniquely combine the benefits of macroscale and nanoscale mechanical resonators and open a new regime for acoustic technologies. These suspended structures will be characterized by their ultrahigh room-temperature Q factors, their ability to firmly integrate with microchip architectures, and their relatively large masses and surface areas. Such centimeter-scale surface areas and masses which are mechanically well-isolated are well suited for high precision measurements of acceleration⁴¹, pressure⁴², and vacuum⁴³. These attributes also make them promising for the observation of mesoscopic quantum behavior in ambient temperatures which are largely limited by room-temperature thermomechanical noise⁴⁴. Reaching a quantum-limited motion regime at room temperature will extend the application range of quantum sensors¹³, quantum calibrated sensing⁴⁵ and quantum memories⁴⁶. The drive towards multi-centimeter nanostructures also holds the potential for applications in ultrasensitive nanomechanical detection⁴⁷, including searches for dark-matter⁴⁸, Casimir forces^{49,50}, and studies of entropy and time⁵¹.*

4. ‘The Q factor of the 3 cm design was a hundred times larger on average, but the optimal designs for both scales differed (Fig. 2b).’ Looking at Fig. 2(b), I can see that the ratio of the quality factors is not uniform, but I cannot assess directly how the optimal designs differ. The sentence would in fact indicate that the scaling method between short and long designs does not work well. Can the authors comment on this apparent contradiction?

We understand that this explanation should be clarified. During the optimization using MFBO, we use the 3 mm and 3 cm designs for simulation but the algorithm aims to optimize the 3 cm model only. That means we do not have a 3 mm optimized model for comparison. The sentence mentioning ‘optimal designs for both differed’ has been updated to ensure everything is clear. We have revised the description of Fig. 2b with a more detailed explanation and removed the 3 mm subset figure in Fig. 2c to avoid confusion.

(line 244-255) *... confirming the expected correlation between the two models according to the soft clamped Q factor in Equation 1. The variation in Figure 2b underscores that what improves performance for a 3 mm resonator does not necessarily translate into a better 3 cm resonator. Therefore, exclusively relying on the low fidelity model (3 mm resonator) would not yield the desired outcome for designing the 3 cm resonator. Simultaneously, despite each resonator size having its unique set of optimal geometric parameters, the algorithm is still capable of learning enough from the response of the 3 mm resonator to guide the optimization of the 3 cm one.*

5. The central theme of the paper is the application of machine learning on two scales to find a design for a long, tapered beam. The result, as the authors write, is similar to the one of Ghadimi et al from 2018. The obvious question to be asked at this point is whether simply making the beam from the EPFL group longer by a factor 4 would have resulted in a similar quality factor $Q > 10^9$, or whether the algorithm provided a better value that human-driven design would have missed.

We thank the reviewer for raising this pertinent question regarding our optimized design versus a scaled version of the EPFL group’s design.

Firstly, we remember that a design optimized for a short resonator does not necessarily scale linearly to a longer resonator (as discussed in the response for Point 4 of reviewer 2) and that the EPFL beam design is also an optimized structure. Extending the EPFL design could potentially achieve a quality factor in the billions, as indicated by the quadratic increase in Q with length shown in Fig. 2b. Specifically, a direct scale-up of the EPFL beam to four times its length could theoretically yield a Q around 6 billion (based on the formula $800 \text{ million} \cdot (3 \text{ cm}/0.7 \text{ cm})^2 / (50 \text{ nm}/20 \text{ nm}) \approx 6 \text{ billion}$) in a rough estimate. The difference in the number of unit cells and other design aspects limits the direct comparison with this study’s result.

More importantly, our work demonstrates that the use of Multi-Fidelity Bayesian Optimization (MFBO) to optimize resonator designs across different scales yields superior results compared to scaling existing designs or single-scale optimization. The MFBO approach efficiently navigates the trade-offs and interactions between different design parameters that a simple scale-up would not capture. We have elaborated on this point and the advantages of our optimization method in the revised manuscript.

(line 260-265) *Even though increasing the length of the resonator has a trend towards increasing the resonator's quality factor, optimization of the phononic crystal parameters is required for each given specific length scale. The advantage of using multi-fidelity transformed Bayesian optimization is shown in Fig. 2d, as it finds a design with Q ...*

In addition, we added a discussion about the optimized design in the revised manuscript. The optimal design recovers the general concept of the physically motivated design of [30], even though the details in the defect's width differed due to the difference in vibration modes (symmetric vs. anti-symmetric). Quantitatively comparing the high Q factor from different mechanisms is tricky since the algorithm was not forced to search the design for each symmetric and anti-symmetric mode. However, we added a more detailed discussion to describe the physics of having a wider defect width in the anti-symmetric optimized mode and clearly stated the optimum design. We have specified our claim with a more detailed explanation as follows.

(line 271-281) *The optimal design maximized the unit cell width around the clamping region and narrowed down the width of the unit cells coming near the center of the resonator. One distinguishing aspect of our approach was that during the optimization under the design domain, the algorithm considered both symmetric and anti-symmetric modes, but the optimized mode was selected to be anti-symmetric rather than symmetric. Because of this difference, the defect width was not minimized, unlike the early study³³, since the kinetic energy with significant movement is not maximized at the center.*

6. How realistic is the expected thermomechanical-limited frequency stability? Have the authors measured the frequency fluctuations at these comparably low resonance frequencies?

Thank you for your question and interest in Allan deviation. First, we mistakenly referred to Allan deviation in the manuscript while calculating the Allan variance. The updated version of the manuscript now contains the correct calculation for the nondimensional Allan deviation. We did not conduct experimental measurements for this specific purpose. However, previous research⁹ conducted on square silicon nitride with comparable resonance frequency (188 kHz) was able to investigate the effect of thermomechanical noise on the Allan deviation in close-loop in both linear and non-linear regime. Therefore, we expect the predicted value of the thermomechanical-limited frequency stability to be experimentally measurable. We have also updated the conclusion of the manuscript to better describe the Allan deviation and how realistic are our assumptions following the reviewer's comment:

(line 448-451) *Conservatively assuming sub-micrometers amplitude displacements in the linear regime, we can extrapolate a thermomechanical limited Allan deviation⁵⁸ $\sigma_y(\tau) \sim 3 \times 10^{-12} / \sqrt{\tau}$ for $m_{eff} = 4.96 \times 10^{-13}$ kg.*

⁹Tomás Manzaneque, Murali K. Ghatkesar, Farbod Alijani, Minxing Xu, Richard A. Norte, and Peter G. Steeneken Phys. Rev. Applied 19, 054074 – Published 23 May 2023

REVIEWER COMMENTS

Reviewer #1 (Remarks to the Author):

In my previous review of “Centimeter Scale Nanomechanical Resonators With Low Dissipation” , I expressed admiration for the reported results on the grounds that

- a. Fabricating centimeter-scale nanomechanical resonators is extraordinarily difficult.
- b. The reported Q factor appears to be the highest reported in the literature
- c. The numerical optimization discussion is timely.
- d. The devices may find application in state of the art fundamental experiments.

I also expressed concerns about publishing the manuscript, on three major grounds

- e. The reported device aspect ratios (the relevant quantity for dissipation dilution) and Q factors are less than a factor of 2 better than prior results, and thus arguably incremental.
- f. Only a single device was reported.
- g. The manuscript makes excessive use of hyperbole, at the expense of clarity and accuracy.

and on twenty-one less major grounds.

The authors have provided substantial responses to my comments and those of the other referee (# 2), including a comprehensive revision of the manuscript and addition of new data in the appendix. To my major points, their response was, in brief,

- e. The authors emphasize that their devices are 4 times larger than previous, and that this represents a nonlinear advance in time (Fig. 1 of their rebuttal). They also modified their discussion, including Eq. 1, to (1) emphasize the difference between increasing size and increasing aspect ratio, from the standpoint of dissipation engineering, (2) better contextualize their reported Q factor and (3) lay more emphasis on the novelty of their fabrication technique.
- f. The authors added three new devices in the appendix, with similar performance.
- g. The manuscript was substantially revised to reduce hyperbole.

Overall, I think these responses---including responses to the 21 less-major points---trump my remaining concerns about the manuscript, and that the decision to publish may be made on its qualitative merits. In this regard, for the points raised above (a-d), and because of the overall high quality of data presentation, I believe the manuscript is appropriate for Nature Communications.

With regards to my remaining concerns, below please find a potpourri of comments re: the authors'. Letters (a-g) and numbers (1-21) refer to my initial comments and the authors' rebuttal. **Comments with red stars refer to misleading or incorrect statements that the authors should strongly consider revising.**

- e. The new equation 1 does a better job of emphasizing the difference between changing aspect ratio and changing size (L). I agree it's an important distinction.

With regards to “the struggle to surpass the centimeter scale over the last decade”, I would say that there hasn't been a struggle, because there hasn't been a real impetus (as Fig. 1 arguably suggests). That doesn't mean it's not interesting though. In general---and in line with the authors' commendable rebuttal to ref #2 comment #3--- it seems advisable to focus on the outlook rather than the struggle, when embarking on adventures like the authors have. Unfortunately, NPJ forces us into a box by insisting on “however” statements in their abstracts...

The notion that cryogenic and/or UHV operation is more “specialized” than the reported fabrication technique is poorly motivated. There are reasons to operate in a cryostat *besides* high Q, and the practical implications of moving from 10^{-9} to 10^{-11} mbar are irrelevant to most practical applications, since vacuum packaging is usually restricted to 10^{-6} mbar. What's

more important/interesting in this reviewers' opinion---and still not explained in the current manuscript--is *why* levitated particles require lower pressure. See comments 6 and 21.

- f. The new data is a valuable (and understandably challenging) addition.
- g. Thanks for humoring me.
- 1*. I personally still find the author's analogy misleading, since the dissipation dilution of a pendulum can be expressed as a ratio of the gravitational stiffness of the pendulum bob ($\sim mg$) to the elastic stiffness of the suspension. If the authors insist on attributing dissipation dilution to tensile stress in the renormalized, cantilever-like suspension mode, then they should cite [19], which argues that these two interpretations are equivalent (and discusses their controversy).
2. Consider modifying line 64-66 to reflect the fact that nanomechanical resonators *by definition* have sub-micron thicknesses, and *by convention* have sub-mm length scales.
- 3*. I feel there's a lost opportunity here by the author's (who are expert in this business) to articulate precisely why Si₃N₄ is so popular. It's *intrinsic* optical and mechanical properties (at least at room temperature) are no better than other standard photonics materials like SiO₂. What sets Si₃N₄ apart is that it is easy to realize strained Si₃N₄ on Si (a minor thermal mismatch miracle?), which allows for the *extrinsic* effect of dissipation dilution to masquerade as low mechanical dissipation. (The material Q is not changed, just the resonator Q).
4. Looks good.
5. As best I can tell from the author's rebuttal to referee #2, they get only a marginal improvement from their numerical optimization. (In other words, they could have made the identical device as the EPFL group, and the Q would have been 6 million rather than 10 million). So I don't buy the need for it *per se*. However, I completely agree with the authors that *in general* one would rather not be constrained to physical models, if the goal is to explore arbitrary resonator geometries at the centimeter-scale. Insofar as they've promoted this perspective in the revised manuscript---rebuttal to ref #2, com #5--- I'm satisfied. Also, as mentioned previously, numerical optimization is a popular subject these days, so the authors are making valuable connections.
- 6*. I find the comparison to vacuum pressures for levitated systems to be misleading, possibly incorrect. Levitated systems don't require lower vacuum to achieve the same Q as nanomechanical resonators *per se*. It depends on the radius of the levitated particle (a sphere, say), the thickness of the nanomechanical resonator, and whether or not they are constrained to have the same vibrational frequency. For example, the $Q = 10^{10}$ trapped particle in [52] has a damping rate of 80 nHz at a pressure of $7 \cdot 10^{-11}$ mbar, which translated to the >100 kHz string modes in this paper would correspond to $Q > 10^{12}$! However, the comparison is unfair, because the trapped particle in [52] has a vibrational frequency of only 1 kHz. If the trap potential were increased to 100 kHz, then (for the same pressure) the two would be on equal footing... In comparing these two different systems on equal footing, the essential fact is that the *damping rate* (independent of Q) of a nanomechanical resonator depends on its thickness, as the authors point out in their Supp. Info, Eq. S3. If the authors insist on comparing gas damping of levitated and nanomechanical systems, then it seems prudent to qualify their assumptions.
7. Line 170-175: Radius of curvature (of what?) not defined. Consider referring to SI (if applicable) or citing a paper.
8. Looks good.

9. I agree that Eq. 1 applies to PnC strings with L replaced by $L_{\text{defect}} = L/n$, where n is the effective mode order. In the EPFL paper this statement is made explicitly. The author's compromise by saying "Eq. 1 doesn't apply directly to PnC strings, but gives insight into general trends with h/L ". While this is not incorrect, a more precise statement would be that it applies, but with $n \gg 1$.
10. See 7.
11. Looks good. (See also 5.) If I had one gripe, it's that it seems that by "higher aspect ratios demand advanced design strategies," the authors mean "demand advanced *simulation* strategies". I.e., the best design is what it is, but the numerical search would have converged to it slowly, if one was not careful. Granted I may not be up on the jargon in this domain.
12. Looks good.
13. Looks good.
14. Looks good.
15. Looks good.
- 16*. Fair enough, however, I still think the authors should remind the reader near line 365 that the Q they measure, $6.6e6$, is lower than the Q they predict, $10e6$, by 30%. (*This is pretty good!*) Then they can refer to speculation about device thickness in the SI. I belabor this because, well, it's the punchline of the "low dissipation" section of the paper, and seems odd to omit.
- 17*. This is the highest Q factor at room temperature, not the lowest damping rate. For example, a 14 microHertz damping rate was measured in [36], for a 35 Hz, $Q = 2.5e6$ resonator.
18. Looks good. (That was indeed my PDF viewer, thanks; maybe revert to compressed vector graphics in lieu of raster, for publication.)
19. Looks good. In the context of *measurement-based* ground state cooling, consider pointing out the large zero-point spectral density of the device, which scales as $Q/mf^2 \sim L^2/(L/L) \sim L^2$.
20. Looks good. However, note (addressing ref #2, com #6) that a fractional AD of $1e-12$ at 100 kHz is not something to consider lightly. The best fractional AD that has been observed in the literature for a nanomechanical resonator is $\sim 5e-10$ (reference [62]), coincidentally for a Si₃N₄ nanostring with a similar frequency (200 kHz) but lower Q , $\sim 10^6$. There are many subtle technical reasons why it is unlikely that an AD of $1e-12$ will follow directly from increasing Q (e.g. spectral diffusion, and increased SNR requirements to observe the sub-damping-time behavior in [62]). The authors don't claim that it will be easy in the manuscript, so this is all fine---I'm just pointing out that their response to ref #2 might require more nuance, in a public venue.
- 21*. (See also comment 6). I'm not sure this argument is valid. The fact that gas damping is sub-dominant at $P < 1e-9$ is simply because the authors have chosen a particular aspect ratio. As the authors point out in their rebuttal--and as explained in comment 6, above---a fair comparison between levitated particles and nanomechanical resonators would require them to have the same frequencies. In this case, it's not clear that one would be better than the other.

I would also point at that $1e-9$ mbar is *not* typically achieved with a turbo pump, but rather with ion pump in a well-conditioned (but not overly complex) SS vacuum chamber. Most AMO physicists will immediately recognize this as an exaggeration. For this audience, it seems prudent to tone down statements like "complex architecture" and "in stark contrast".

Reviewer #2 (Remarks to the Author):

The authors have replied to most of my comments very well and extensively. I am still not convinced that devices in the mm and cm scales should be considered as different classes of devices – this would imply a qualitative difference between the two length scales, which I find hard to understand. Nevertheless, this paper has many strong points and fulfills the criteria for being published in Nature Communications without further changes.

Responses to the comments of reviewers
Centimeter-scale nanomechanical resonators with low dissipation
reference number No. NCOMMS-23-40247A

We thank the editor and reviewers for their enthusiasm about our results and for their feedback. Reviewers' comments are in **bold**, responses to the reviewers are in black, and added or modified sentences are in *italic and magenta*. To facilitate a more straightforward identification of each modification made to the manuscript, we have introduced line numbers throughout the document to assist reviewers in pinpointing and referencing specific changes easily. Reviewer 2 supports the publication of the manuscript without further modifications. Therefore, this response letter addresses the remarks of Reviewer 1.

Comments from Reviewer #1:

In my previous review of “Centimeter Scale Nanomechanical Resonators With Low Dissipation”, I expressed admiration for the reported results on the grounds that

- a. Fabricating centimeter-scale nanomechanical resonators is extraordinarily difficult.
- b. The reported Q factor appears to be the highest reported in the literature.
- c. The numerical optimization discussion is timely.
- d. The devices may find application in state of the art fundamental experiments.

I also expressed concerns about publishing the manuscript, on three major grounds.

- e. The reported device aspect ratios (the relevant quantity for dissipation dilution) and Q factors are less than a factor of 2 better than prior results, and thus arguably incremental.
- g. Only a single device was reported.
- f. The manuscript makes excessive use of hyperbole, at the expense of clarity and accuracy.

and on twenty-one less major grounds.

The authors have provided substantial responses to my comments and those of the other referee (# 2), including a comprehensive revision of the manuscript and addition of new data in the appendix. To my major points, their response was, in brief,

- e. The authors emphasize that their devices are 4 times larger than previous, and that this represents a nonlinear advance in time (Fig. 1 of their rebuttal). They also modified their discussion, including Eq. 1, to (1) emphasize the difference between increasing size and increasing aspect ratio, from the standpoint of dissipation engineering, (2) better contextualize their reported Q factor and (3) lay more emphasis on the novelty of their fabrication technique.
- g. The authors added three new devices in the appendix, with similar performance.
- f. The manuscript was substantially revised to reduce hyperbole.

Overall, I think these responses—including responses to the 21 less-major points—trump my remaining concerns about the manuscript, and that the decision to publish may be made on its qualitative merits. In this regard, for the points raised above (a-d), and because of the overall high quality of data presentation, I believe the manuscript is appropriate for Nature Communications.

With regards to my remaining concerns, below please find a potpourri of comments re: the authors'. Letters (a-g) and numbers (1-21) refer to my initial comments and the authors' rebuttal. Comments with red stars refer to misleading or incorrect statements that the authors should strongly consider revising.

Frankly, we want to thank this reviewer for the constructive feedback and extensive review of our work, including the few additional comments that were highlighted with red stars. We are also glad to hear that this reviewer also believes the manuscript is appropriate for Nature Communications, in agreement with the other reviewer.

e. The new equation 1 does a better job of emphasizing the difference between changing aspect ratio and changing size (L). I agree it's an important distinction. With regards to "the struggle to surpass the centimeter scale over the last decade", I would say that there hasn't been a struggle, because there hasn't been a real impetus (as Fig. 1 arguably suggests). That doesn't mean it's not interesting though. In general—and in line with the authors' commendable rebuttal to ref #2 comment #3— it seems advisable to focus on the outlook rather than the struggle, when embarking on adventures like the authors have. Unfortunately, NPJ forces us into a box by insisting on "however" statements in their abstracts... The notion that cryogenic and/or UHV operation is more "specialized" than the reported fabrication technique is poorly motivated. There are reasons to operate in a cryostat besides high Q, and the practical implications of moving from 10^{-9} to 10^{-11} mbar are irrelevant to most practical applications, since vacuum packaging is usually restricted to 10^{-6} mbar. What's more important/interesting in this reviewers' opinion—and still not explained in the current manuscript—is why levitated particles require lower pressure. See comments 6 and 21.

Thank you for this comment. The responses to comments 6 and 21 are included below.

- f. The new data is a valuable (and understandably challenging) addition.
- g. Thanks for humoring me.

1*. I personally still find the author's analogy misleading, since the dissipation dilution of a pendulum can be expressed as a ratio of the gravitational stiffness of the pendulum bob ($\sim mg$) to the elastic stiffness of the suspension. If the authors insist on attributing dissipation dilution to tensile stress in the renormalized, cantilever-like suspension mode, then they should cite [19], which argues that these two interpretations are equivalent (and discusses their controversy).

Indeed, eq. (4) of reference [19] results from the derivation of the dissipation dilution factor of a pendulum, which only depends on the geometry of the wire, its Young's modulus, and the applied tension. The main difference between dissipation dilution in a pendulum and a string resonator is in the way that tension is generated by gravitational force or by fabrication-induced prestress. We followed the reviewer's suggestion and added the following sentence to the article:

(new lines 42-45) *The reader is referred to Cagnoli et al.¹⁹ for a discussion about the use of 'dissipation dilution' terminology for pendula and prestressed nanomechanical resonators.*

2. Consider modifying line 64-66 to reflect the fact that nanomechanical resonators by definition have sub-micron thicknesses, and by convention have sub-mm length scales.

Thank you. We have updated the main text to provide more precise definitions for the terms used.

(new lines 67-70) *At the nanoscale, nanomechanical resonators possess significantly reduced thickness, in the range of sub-micrometers, with a length limited to a sub-millimeter length by convention.*

3*. I feel there's a lost opportunity here by the author's (who are expert in this business) to articulate precisely why Si₃N₄ is so popular. It's intrinsic optical and mechanical properties (at least at room temperature) are no better than other standard photonics materials like SiO₂. What sets Si₃N₄ apart is that it is easy to realize strained Si₃N₄ on Si (a minor thermal mismatch miracle?), which allows for the extrinsic effect of dissipation dilution to masquerade as low mechanical dissipation. (The material Q is not changed, just the resonator Q).

Thank you for your comment. We have expanded in the main text on the advantages of using silicon nitride as detailed below:

(new lines 79-82) ... and specific advantages gained through strained configurations on silicon, emphasizing the role of strain engineering for dissipation characteristics.

4. Looks good.

5. As best I can tell from the author's rebutal to referee #2, they get only a marginal improvement from their numerical optimization. (In other words, they could have made the identical device as the EPFL group, and the Q would have been 6 million rather than 10 million). So I don't buy the need for it per se. However, I completely agree with the authors that in general one would rather not be constrained to physical models, if the goal is to explore arbitrary resonator geometries at the centimeter-scale. Insofar as they've promoted this perspective in the revised manuscript—rebutal to ref #2, com #5—I'm satisfied. Also, as mentioned previously, numerical optimization is a popular subject these days, so the authors are making valuable connections.

Indeed, this was the main motivation of exploring the multi-fidelity optimization strategy. We are glad we are aligned.

6*. I find the comparison to vacuum pressures for levitated systems to be misleading, possibly incorrect. Levitated systems don't require lower vacuum to achieve the same Q as nanomechanical resonators per se. It depends on the radius of the levitated particle (a sphere, say), the thickness of the nanomechanical resonator, and whether or not they are constrained to have the same vibrational frequency. For example, the $Q = 10^{10}$ trapped particle in [52] has a damping rate of 80 nHz at a pressure of $7 \cdot 10^{-11}$ mbar, which translated to the >100 kHz string modes in this paper would correspond to $Q > 10^{12}$! However, the comparison is unfair, because the trapped particle in [52] has a vibrational frequency of only 1 kHz. If the trap potential were increased to 100 kHz, then (for the same pressure) the two would be on equal footing... In comparing these two different systems on equal footing, the essential fact is that the damping rate (independent of Q) of a nanomechanical resonator depends on its thickness, as the authors point out in their Supp. Info, Eq. S3. If the authors insist on comparing gas damping of levitated and nanomechanical systems, then it seems prudent to qualify their assumptions.

This is a pertinent point. Our purpose was not to provide an unfair comparison. Instead, we just wanted to compare the Q of the presented device with the best Q we have found in the literature for levitated systems up to now. Therefore, we provide additional clarification in the article that these two devices operate at different frequencies. We also replace 'dissipation levels' with 'quality factor' to highlight that the comparison involves quality factors rather than damping rates. As a short note, the reviewer argues that if it is possible to raise the stiffness of the optical trap potential by a factor of 100, then the comparison of the two systems would be on equal footing. Increasing the resonance frequency of a levitated particle by a factor of 100 at the same mass and damping coefficient c is currently still very difficult considering the experimental state-of-the-art, and raising the frequency by a factor of 100, requires increasing the stiffness by a factor of 10000, which does not seem trivial at all. The question is whether or not this would lead to excessive optothermal heating (melting) of the particle or other effects that reduce the Q of the system. Nevertheless, we prefer to avoid conjecturing about this in our article, as these questions are outside the scope of the current work.

(new lines 151-152) *In contrast, our clamped centimeter-scale resonators are limited by intrinsic losses²⁰ in view of their higher vibrational frequency, and can approach comparable quality factors ...*

7. Line 170-175: Radius of curvature (of what?) not defined. Consider referring to SI (if applicable) or citing a paper.

We have added in the main text that the curvature is defined for the resonator's bending deformation:

(new lines 184-185) *The curvature is defined as the reciprocal of the radius of curvature for the resonator's bending deformation.*

8. Looks good.

9. I agree that Eq. 1 applies to PnC strings with L replaced by $L_{defect} = L/n$, where n is the effective mode order. In the EPFL paper this statement is made explicitly. The author's

compromise by saying “Eq. 1 doesn’t apply directly to PnC strings, but gives insight into general trends with h/L ”. While this is not incorrect, a more precise statement would be that it applies, but with $n \gg 1$.

Equation 1 is the general form of the quality factor derivation over a vibrating pre-stressed string, which is still valid even when we increase the mode number (note that the equation contains the mode number in the formulation itself). We highlight that the equation is not only predicting the defect’s dissipation, but it actually predicts the loss over the entire resonator. In other words, we do not update the defect length as L/n to evaluate the quality factor. However, the reviewer’s comment illustrates that our original sentences were not completely clear, so we updated them accordingly:

(line 167-171) *Note that this equation assumes perfect clamping to the substrate without considering the mechanical coupling, potentially affecting the Q factor at a smaller mode order⁵⁵.*

(line 197-200) (...) *Equation 1 is a straight beam formulation that does not directly apply to phononic crystal strings⁴⁹ (due to the geometry difference) but can give insight into general trends with h/L .*

10. See 7.

11. Looks good. (See also 5.) If I had one gripe, it’s that it seems that by “higher aspect ratios demand advanced designs strategies,” the authors mean “demand advanced simulation strategies”. I.e., the best design is what it is, but the numerical search would have converged to it slowly, if one was not careful. Granted I may not be up on the jargon in this domain.

This is also a fair point. We changed ‘*design strategies*’ to ‘*simulation-based design strategies*’ to be more precise.

(new lines 209-210) ... *demand advanced simulation-based design strategies ...*

12. Looks good.

13. Looks good.

14. Looks good.

15. Looks good.

16*. Fair enough, however, I still think the authors should remind the reader near line 365 that the Q they measure, 6.6e6, is lower than the Q they predict, 10e6, by 30%. (This is pretty good!) Then they can refer to speculation about device thickness in the SI. I belabor this because, well, it’s the punchline of the “low dissipation” section of the paper, and seems odd to omit.

In the updated manuscript, we mentioned this point more clearly in the ‘*Centimeter scale nanofabrication*’ section and ‘*Low dissipation at room temperature*,’ where we previously referred to SI section B.

(new lines 363-366) *This is caused by the difficulty of dissipating heat during the etching step of the silicon substrate, which is expected to decrease the Q factor predicted from the simulation (Supplementary Information Sec. B).*

(new lines 419-421) *thus reducing the fidelity between design and experiment with a lower measured Q (Supplementary Information Sec. B).*

17*. This is the highest Q factor at room temperature, not the lowest damping rate. For example, a 14 microHertz damping rate was measured in [36], for a 35 Hz, $Q = 2.5e6$ resonator.

Thank you for the comment. Here is the appropriate update:

(line 436) ... *the highest Q factor yet measured ...*

18. Looks good. (That was indeed my PDF viewer, thanks; maybe revert to compressed vector graphics in lieu of raster, for publication.)

19. Looks good. In the context of measurement-based ground state cooling, consider pointing out the large zero-point spectral density of the device, which scales as $Q/mf^2 \sim L^2/(L/L) \sim L^2$.

20. Looks good. However, note (addressing ref #2, com #6) that a fractional AD of 1e-12 at 100 kHz is not something to consider lightly. The best fractional AD that has been observed in the literature for a nanomechanical resonator is 5e-10 (reference [62]), coincidentally for a Si3N4 nanostring with a similar frequency (200 kHz) but lower Q, 10^6 . There are many subtle technical reasons why it is unlikely that an AD of 1e-12 will follow directly from increasing Q (e.g. spectral diffusion, and increased SNR requirements to observe the sub-damping-time behavior in [62]). The authors don't claim that it will be easy in the manuscript, so this is all fine—I'm just pointing out that their response to ref #2 might require more nuance, in a public venue.

We appreciate the comment.

21*. (See also comment 6). I'm not sure this argument is valid. The fact that gas damping is subdominant at $P < 1e-9$ is simply because the authors have chosen a particular aspect ratio. As the authors point out in their rebuttal—and as explained in comment 6, above—a fair comparison between levitated particles and nanomechanical resonators would require them to have the same frequencies. In this case, it's not clear that one would be better than the other. I would also point out that 1e-9 mbar is not typically achieved with a turbo pump, but rather with ion pump in a well-conditioned (but not overly complex) SS vacuum chamber. Most AMO physicists will immediately recognize this as an exaggeration. For this audience, it seems prudent to tone down statements like “complex architecture” and “in stark contrast”.

We agree with the reviewer that the gas damping force of a structure at a certain pressure mainly depends on the resonator geometry. A linear damping force is characterized by a damping coefficient c and a velocity proportional damping force $F_d = -cv$. For a fixed geometry and pressure, this coefficient c is the same for a levitating resonator and a string resonator. To emphasize it, we have highlighted in the manuscript that string resonators and levitated particles operate at different vibration frequencies.

As we detailed in our response to comment 6, the quality factor also depends on the stiffness and mass of the structure; according to $Q = \sqrt{km}/c$, Q can be increased by increasing the stiffness. In our view, the stiffness k that can be realized in a high-stress string is significantly higher than the stiffness that can be realized in levitating traps with any currently known technology at comparable device geometries. We do not exclude that in the future, it will become possible to realize levitating traps with higher stiffness, but considering the experimental state-of-the-art, this is currently still very difficult.

To attain 1e-9 mbar, we employ a turbo pump with a final pressure of 5×10^{-10} mbar (Pfeiffer Turbo HiPace 80N 63 CF-F) and attach it directly to the chamber through a large opening (CF62) without any restrictive/long hoses that limit the turbo's ability to pump the chamber. Indeed, if one uses a typical turbo-pump station where a chamber is connected to the pump via vacuum hoses, it is very difficult to get 1e-9 mbar. We do not have a load-lock; thus, we fully open the chamber for every new sample, and never need to bake our chamber. Moreover, we employed only UHV-compatible elements inside the chamber including lead-free soldering, copper gaskets and KAPTON wires, and the chamber is purged with nitrogen 3 consecutive times before turning on the turbo pump at every measurement (further details on the setup can be found in Appendix D of the Ph.D. thesis Cupertino, A. "Exploring the quality factor limits of room temperature nanomechanical resonators." (2023)).

The pressure is measured at the chamber by a Pfeiffer PKR 360 gauge (range 10^{-9} – 10^3 mbar) employing Pirani and Cold cathode sensors. Given the final pressure of the turbo pump and the accuracy of the pressure gauge (30%), the read-out pressure only gives a lower estimate of the vacuum. It could be that perhaps our vacuum is, in fact, not actually as low as 1e-9 mbar but closer to 1e-8 mbar, but this would actually be a positive claim for our platform at more ambient pressures. We do not want to make such a bold claim, so use the more conservative limit of 1e-9 mbar. We have added these details in the SI so it is clear that these low pressures are possible due to the turbo's proximity and avoidance of interconnects between the pump and chamber, and toned down the relevant statements between lines 468 and 477.

(line 468-477) *This comparison gains further importance when considering that levitated particles are essentially isolated from the environment, interacting only minimally with residual gas molecules at vacuum levels as low as 10^{-11} mbar, which requires infrastructures different from the turbo and roughing pump combination used in this study (Supplementary Information Sec. K). In contrast, our solid-state resonators are physically clamped to a room-temperature chip, surrounded by 100 times higher gas pressures and exhibit comparable acoustic dissipation.*

(New section: Supplementary Information K)

Ultra High Vacuum Setup

We developed a UHV setup designed to operate at pressures close to 10^{-9} mbar, aiming to minimize gas damping and measure the intrinsic quality factor of the fabricated centimeter-scale resonators. This pressure value approaches the final pressure of the employed turbomolecular pump, necessitating setup optimization to reduce the volume and the surface of the overall vacuum system, and potential outgassing sources. This section provides details on the vacuum setup.

The vacuum setup consists of a cylindrical chamber (Fig. S13a) with 8 CF-40 flanges along the lateral surface and 2 CF-100 flanges on the top and the bottom. The CF-40 flanges are employed for electrical and optical connection and the vacuum gauge. The bottom flange directly connects to a turbo molecular pump via a valve and a large opening CF-63 (Fig. S13b) without any restrictive hoses that could hinder pumping efficiency. This allows the volume and the surface of the overall vacuum system to be reduced, improving the pumping efficiency. The chamber incorporates UHV-compatible components for sample manipulation, including a triaxial nanopositioner, sample holders, and piezoelectric plates soldered with lead-free solder to KAPTON wires. Lacking a load lock, we fully open the chamber for each new sample through the top viewport and do not require chamber baking. After loading a sample, we typically purge the chamber with nitrogen three times while the backing pump is running before starting the turbo molecular pump.

It is crucial to note that the employed pressure gauge's operational range spans from 10^{-9} to 10^3 mbar, whereas the turbomolecular pump's final pressure is 5×10^{-10} mbar. Consequently, the pressure gauge provides only a lower estimate of the vacuum level.

Fig. S13: Vacuum chamber and turbo molecular pump. **a**, Photograph of the vacuum chamber with the pressure gauge mounted at the vacuum chamber. **b**, Photograph of the turbo molecular pump connected to the bottom of the vacuum chamber.

REVIEWERS' COMMENTS

Reviewer #1 (Remarks to the Author):

The authors have (painstakingly) addressed all of my comments. I appreciate their consideration and support their manuscript for publication.